# HardTestGen: A High-Quality RL Verifier Generation Pipeline for LLM Algorithmic Coding

**Zhongmou He**[1*] **Yee Man Choi**[1*] **Kexun Zhang**[1*†] **Jiabao Ji**[2]
**Junting Zhou**[1] **Dejia Xu**[3] **Ivan Bercovich**[2] **Aidan Zhang**[1]
**Yixiao Zeng**[14] **Lei Li**[1]
[1]Carnegie Mellon University  [2]UC Santa Barbara  [3]UT Austin  [4] Meracode

## Abstract

Verifiers provide important reward signals for reinforcement learning of large language models (LLMs). However, it is challenging to develop or create reliable verifiers, especially for code generation tasks. A well-disguised wrong solution program may only be detected by carefully human-written edge cases that are difficult to synthesize automatically. To address this issue, we propose HardTestGen, an approach to synthesize high-quality test cases for algorithmic coding problems. We curate a comprehensive algorithmic programming dataset HardTests with 26.6k problems and high-quality synthetic tests. Compared with existing tests, HardTestGen tests demonstrate significantly higher accuracy in verifying LLM-generated code (+11.22 percentage points in precision, the percentage of actually correct code within the predicted correct ones). We also show that downstream post-training — including rejection sampling and reinforcement learning (RL) — using HardTests verifier results in improved performance of LLM code generation. We open-source our dataset and synthesis pipeline at `https://leililab.github.io/HardTests/`.

## 1 Introduction

Post-training large language models (LLMs) with outcome verifiers[1] (Guo et al., 2025; Kimi Team et al., 2025) can greatly improve their reasoning ability. LLMs trained with these techniques are approaching the level of the best humans on challenging problems in math and programming olympiads (OpenAI et al., 2025). To properly assign outcome rewards in post-training, reliable verifiers are needed for both reinforcement learning and rejection sampling.

For coding, verifiers are often test cases (Le et al., 2022; Singh et al., 2023) that tell right algorithms from wrong ones. Algorithmic coding requires efficient solutions with advanced data structures and algorithms. The ability to solve these problems is essential for efficiency-critical domains such as high-performance computing, but its complex nature poses challenges for obtaining accurate verifiers and LLM reinforcement learning. A bad choice of algorithm can lead to a well-disguised wrong solution, which may easily pass random tests but still break on human-written special cases. Consider this example problem: *for a rooted tree with $n$ nodes and weighted edges, calculate the sum of path lengths from every node to the root node.* A naive algorithm that enumerates all such paths and sums edge by edge has a time complexity of $\Theta(nd)$, where $d$ is the depth of the tree. This can be decently efficient in many cases, as $\mathbb{E}[d] = \Theta(\log n)$ for randomly generated trees (Devroye et al., 2012). For such an algorithm to time out, the test case needs to be a valid tree that is large enough (so that $n$ is large) and special enough (so that $d$ is large). A chain (each non-leaf node has exactly one child), whose depth $d = n$, can cause the algorithm to be as slow as $\Theta(n^2)$. This example demonstrates the need for valid, comprehensive tests to accurately verify algorithmic coding and assign rewards.

---

*Equal contribution. †Project lead. Correspondence to {zhongmou,yeemanc,kexunz}@andrew.cmu.edu

[1]In this paper, "verifier" refers to rule-based systems that attempt to check the correctness of problem solutions. **"Verifiers" do not necessarily guarantee correctness.**

Generating valid and comprehensive tests is hard. Existing test synthesis methods, such as CodeT (Chen et al., 2023) and TACO (Li et al., 2023), rely on LLMs to directly write test inputs. Consequently, existing datasets of coding problems and associated test cases are less than comprehensive. 60% of the programs that pass test cases in APPS (Hendrycks et al., 2021) are in fact, wrong. 46% of the programs that pass test cases in CodeContests (Li et al., 2022) are semantically correct, but too inefficient to pass human-written tests. More importantly, scraping human-written tests is unfeasible — according to our study, for most of the problems, human-written test cases are proprietary and impossible to scrape, demanding synthesized tests.

To alleviate these issues, we propose HARDTESTGEN, an LLM-based test synthesis pipeline. Our main insights are 1) test cases' validity is better preserved when generated from LLM-produced programs rather than directly from the LLMs themselves, and 2) each test generator has different hypotheses about the programs under test and creates tests from a different distribution. With these insights, HARDTESTGEN establishes a unified pipeline that synthesizes four types of test inputs. Among them, LLMGen is based on direct LLM generation, while the other three types — RPGen, SPGen, and HackGen — are produced by LLM-written generator programs. For outputs, HARDTESTGEN relies on multiple human-written oracle programs to compute expected results and applies consensus filtering to eliminate invalid cases. Such oracle programs are available for the vast majority of problems in online coding competitions.

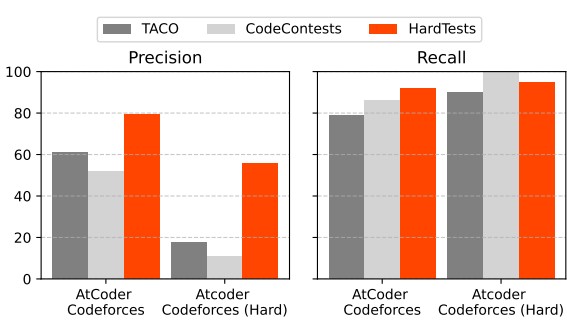

Figure 1: HARDTESTS test cases are significantly better than the baselines. The large improvement in precision indicates that our tests greatly reduce false positives and are indeed *harder*.

With HARDTESTGEN, we curate HARDTESTS, a comprehensive dataset for coding competitions with 26.6k problems and high-quality test cases. As shown in Figure 1, compared to existing test synthesizers, HARDTESTS tests are more reliable in terms of precision and recall when evaluating programs generated by Qwen2.5-Coder-7B (Yang et al., 2024). The gap in precision can be up to 40 percentage points for harder problems. Higher reliability of verification makes HARDTESTS the ideal playground for post-training research in the coding domain. To further demonstrate the benefits of high-quality tests, we conduct post-training experiments with HARDTESTS and baseline tests. Our experiments in 2 different scenarios show that test quality matters significantly for rejection sampling and reinforcement learning. Higher-quality tests can lead to improvements in downstream performance.

In summary, this work provides:

- HARDTESTGEN, an LLM-based test synthesis pipeline that generates high-quality test cases for coding problems, improving precision by 11.23 percentage points and recall by 11.03 percentage points on average.
- HARDTESTS, a comprehensive problem set for competition-level code generation, with 26.6k problems, each with high-quality test cases generated by HARDTESTGEN.
- Empirical analyses on how test quality affects LLM post-training. We show that test quality is of great importance for rejection sampling and reinforcement learning.

## 2 RELATED WORK

**RLVR.** Reinforcement learning has shown great potential in improving LLM reasoning abilities in various domains, such as math (Guo et al., 2025; Zeng et al., 2025b; Ren et al., 2025) and coding (OpenAI, 2025; Liu & Zhang, 2025; Luo et al.). The resulting long-reasoning LLMs, such as OpenAI-o3 (OpenAI, 2024) and DeepSeek-R1 (Guo et al., 2025), largely outperform short-reasoning LLMs through simple RL training to improve outcome-based reward, i.e., whether the model-generated code solution passes all test cases. Although some previous works have explored heuristic rules for selecting training data to improve RL performance (Ye et al., 2025; Wang et al.,

2025b; Li et al., 2025) or reward design (Hou et al., 2025; Kimi Team et al., 2025; Costello et al., 2025), the impact of test case quality on coding LLMs during RL training remains underexplored. In this work, we show that high-quality test cases, i.e., those better at detecting subtle bugs in code, can largely improve coding LLM performance after RL training.

**LLM-based test synthesis.** Test cases are crucial in evaluating the functional correctness and performance of LLM-generated code. Benchmarks such as HumanEval (Chen et al., 2021), and APPS (Hendrycks et al., 2021) provide hand-written test cases that serve as a proxy for code correctness. However, such human-authored test cases are often only publicly available for a limited set of problems. CodeContests (Li et al., 2022) generates additional test cases by mutating existing crawled inputs. Several efforts leverage LLMs by generating test inputs with LLMs and outputs with reference implementation (Li et al., 2023), providing the reference implementation to LLMs to synthesize seed input (Liu et al., 2023), synthesizing test inputs and (pseudo)-oracle programs for test outputs (Chen et al., 2023; Zhang et al., 2023), or even generating coding questions, reference solutions, and tests all with LLMs (Xu et al., 2025; Zeng et al., 2025a). STGen (Peng et al., 2025) generates stressful test cases for evaluating the time efficiency of code. Although existing LLM test synthesis methods prove to be useful in many scenarios, their quality is far from perfect. Concurrently with our work, rStar-Coder (Liu et al., 2025) and HF-Codeforces (Penedo et al., 2025) also study more reliable test synthesis in the competition context. Compared to them, our work highlights a thorough analysis of test quality and a unique set of post-training experiments that demonstrate the downstream effects of high-quality tests. Concurrently with our work, CodeContests+ (Wang et al., 2025c) and Klear-CodeTest (Fu et al., 2025) also explore test-case generation for code reinforcement learning for 12k and 28k problems and study the impact on RL training, respectively. Compared to their work, we also discuss the implications in other training scenarios. We present a more thorough discussion of early test generation approaches, LLM-based test synthesis in the software testing field, quality issues in LLM synthetic tests, and their implication in Appendix A.1.

**Datasets for competition code generation.** Existing datasets for competition code generation focus on scaling the number of problems and CoTs. Luo et al. filters a high-quality 24k problem set of TACO, LiveCodeBench, and other contest programming problems. CodeForces-CoTs, the dataset of 10k Codeforces problems created by Penedo et al. (2025), contains 100k reasoning traces and solutions generated by DeepSeek R1. OpenCodeReasoning (Ahmad et al., 2025) also compiles a dataset of 28k problems, generates 735k reasoning traces, and filters them for syntactic correctness. While these efforts have shown that better models can be trained with more data and more trajectories from teacher models, they are facing a "code verifiability crisis", as described by Open-R1 (Face, 2025), and programs that pass test cases in these problem sets are not necessarily correct. In our paper, we curate HARDTESTS, the large-scale algorithmic coding problem set with 26.6k problems. More importantly, we push the scaling of training data towards higher quality of test cases and evaluate how test quality affects model training.

## 3 THE HARDTESTGEN METHOD

### 3.1 OVERVIEW

We aim to automatically synthesize test cases for algorithmic coding problems that can be used as verifying rewards in code LLM post-training (e.g., reinforcement learning). Given a natural language described algorithmic coding problem $x \in \mathcal{X}$ ($\mathcal{X}$ indicates all possible problems) and a set of correct solution programs $\{y_1^*, y_2^*, \ldots, y_k^*\}$, denoted as "oracle programs", the task of test synthesis is to automatically generate a set of test cases to verify a candidate program $y$'s functional correctness and efficiency. Each set of test cases consists of several inputs, their ground-truth outputs, and an output judging function, which checks the equivalence of candidate outputs and ground-truth outputs. For most cases, the output judging function is a simple string comparison (which is the default). In some rarer cases, we need a special judging function (e.g., set comparison).

We collect a large-scale dataset of algorithmic coding problems from 13 coding competition platforms (e.g., Codeforces). Most of these problems (68%) are accompanied by one or more oracle programs. We filter out the problems without any oracle programs, and those do not read the input from and write the output to the standard I/O.

Figure 2: The procedure of generating test inputs and test outputs in HARDTESTGEN.

Notice that we do not often have access to the golden set of test cases prepared by the creators of these coding problems. Therefore, we cannot directly compare our synthesized test cases against the golden ones. However, we can submit the LLM-generated or human-written candidate program to the source problem's online judge platform to obtain a ground-truth verdict of the solutions' correctness and efficiency. These verdicts can be used to check the correctness of synthesized tests, but they cannot directly be used as reward signals in reinforcement learning since it is extremely time-consuming.

The purpose of our synthetic test cases is to verify the correctness and efficiency of a generated candidate program so that only the programs implementing the right algorithms and data structures would pass all tests. The key challenges are: 1) how can we ensure the input data of a test case is valid, in terms of both content and format? 2) how can we ensure the test case set are comprehensive, covering corner cases and computationally costly ones? 3) how can we obtain ground-truth output results of the problems?

## 3.2 SYNTHESIZING TEST INPUTS FOR ALGORITHMIC CODING PROBLEMS

As illustrated in Figure 2, our HARDTESTGEN includes four techniques to synthesize test case input data: 1) LLMGen — direct LLM generation, 2) RPGen — range-based programmed test input generation where the synthesizer program itself is generated by LLM, 3) SPGen — stratified programmed test input generation according to output value categories, and 4) HackGen: hacking test input generation (i.e., hard cases or edge cases) through specially designed programs.

**LLMGen: direct LLM generation.** We prompt an LLM to directly generate $n_L = 10$ inputs by imitating the sample test cases provided in the problem specification. This type of input is typically small in scale, making it easy to generate and understand, and allowing for quick testing of the candidate program's functional correctness. An example prompt snippet is:

```
Please generate 10 valid inputs according to the problem stated below. You may
follow the examples given in the problem description and generate variations that
are different. Please respect the constraints and data types in the description.
```

**RPGen: range-based programmed test input synthesis using LLM-generated programs.** It is hard for an LLM to generate large-scale, valid inputs. However, it is possible for LLM to identify input data types and ranges. Based on this observation, we prompt an LLM to generate a Python function with no arguments that returns random input data according to the problem's data type, range, and inherent constraints (e.g., x-y coordinates forming a convex polygon). We execute the generator function $n_R = 20$ times to get random test case inputs. An example prompt snippet is:

```
Given the problem described below, please identify the input data types (e.g.,
int, string), ranges, and data constraints (e.g., x-y coordinates forming a
convex polygon or number of items which should be nonnegative), and then generate
a python function "gen_range_based_input" which will return one random input data
for the problem with respecting to these types, ranges, and constraints.
```

**SPGen: stratified programmed test input synthesis according to output value categories.** Coding problems may expect a categorical output value (e.g., Yes or No). Random generation may produce imbalanced test cases — the test cases could always be associated with Yes as ground-truth output — that would not be sufficient to fully verify a candidate solution's correctness. To mitigate such an imbalance, we first prompt the LLM to identify all the output value categories of a problem, and then, for each output value category, we instruct the LLM to write an input-generating function

that only produces inputs associated with such an output category. We execute each input-generating function $n_S = 10$ times to obtain a total of $m_S \times n_S$ inputs, where $m_S$ is the number of categories inferred from the problem by the LLM. An example prompt snippet is:

```
Given the problem described below, please identify how many categories there are
in the output values, denoted as m_S. For each category, please generate one
function "gen_stratified_input_<category_label>" in Python, which can produce
a random input data that will output one value in the given category. Please
replace <category_label> with the inferred output value categories.
```

Note that SPGen can be considered a hierarchical case of RPGen, because we conduct the RPGen process for each output category. Therefore, these two input types are mutually exclusive. For problems that require SPGen, we do not apply RPGen.

**HackGen: hacking test input generation.** There are often candidate programs that are only correct for certain input data — there might be neglected corner scenarios or computationally inefficient for worst-case scenarios. A hacking test case contains the input that will cause a flawed candidate program to either produce an incorrect result or to exceed the running time limit. Previous methods could not generate hacking test cases explicitly. For a given problem, we first prompt the LLM to describe multiple flawed candidate programs using brute-force or some simple classic algorithms such as depth-first search. We then prompt the LLM to think about scenarios where these programs will fail. Then, for each scenario, we prompt the LLM to design an input-generating function to generate inputs corresponding to that scenario. We execute each function $n_H = 10$ times and obtain $m_H \times n_H$ inputs, where $m_H$ is the number of failing scenarios. An example prompt snippet is:

```
Given the problem described below, please first describe several flawed solution
programs using brute-force enumeration or a classic algorithm (e.g., depth-first
search), by explicitly ignoring some constraints, corner conditions, or data
range. Secondly, please think about scenarios where these programs will fail or
exceed the time limit (TLE). Thirdly, for each scenario, generate one function
"gen_hacking_input_<scenario_type>" which will produce possibly random input
data corresponding to such a failing scenario.
```

### 3.3 VALIDATING TEST CASE INPUT DATA USING SYNTHESIZED PROGRAMS

The above synthesized test inputs are not guaranteed to be valid. Instead of directly using LLMs to judge the validity of these test case inputs, we prompt an LLM to generate a function in Python, which takes a generated test case input string as an argument and returns a boolean answer indicating the test input's validity. We specifically instruct the model to check the value types, range, numerical relations, and logical constraints. An example prompt snippet is:

```
Given the problem described below, please identify the value type and range,
and list all constraints on the input data. Then please generate a Python
function "validate_input(input_str: str) -> bool" that checks the input against
all constraints and returns a boolean indicating whether the input is valid
according to the problem description.
```

In our implementation, HARDTESTGEN includes the generated input validator and an oracle solution program together with the four techniques' prompts to generate test inputs, as we find that doing so increases the LLM's likelihood of synthesizing valid inputs and input generators. After generating initial test inputs, we apply the generated validator on these test inputs to eliminate all invalid ones.

### 3.4 COMPUTING EXPECTED OUTPUTS AND FILTERING TEST CASES

HARDTESTGEN generates test outputs using oracle solution programs and applies consensus filtering to retain only reliable test cases. After synthesizing the inputs, we collect up to $n_{\text{oracle}} = 8$ human-written oracle programs for each problem, prioritizing those from more reliable sources. Each oracle program is executed on all synthesized inputs to produce outputs. If two oracle programs generate outputs that are equivalent on more than 90% of the inputs (i.e., semantically the

same rather than strictly identical), we regard this agreement as valid. The synthesized inputs, together with these consensus-equivalent outputs, form the final test cases for this problem.

For most problems, we use direct string comparison to check the equivalence between two programs' outputs. However, for certain problems, this is insufficient. For example, expected output may be a set, where element order does not matter, or a sequence of operations that achieves the desired effect. For these problems, we prompt the LLM to generate a special output judging function, which takes the test input and two outputs as arguments and returns a Boolean indicating whether the outputs are equivalent. In our dataset, 25.4% of the problems require such a special output judging function. In subsequent training and testing processes, this judging function will continue to be used to determine whether the candidate output and the reference output match. An example prompt snippet is:

```
Given the problem described below, please generate an output judging function
in Python to compare the equivalence of two output results. The function takes
the input (can be a list of numbers or strings) and two results as arguments,
and returns a Boolean value. For example, you should use set comparison when
the order of results does not matter.
```

In our dataset, we use GPT-4o to generate all of the above contents if needed. On average, the OpenAI API cost for generating test cases (including inputs and a possible special output judge function) for each problem is 0.23 USD. For all functions that need to be generated, we include two to three carefully crafted examples in the prompts. The implementation details of HARDTESTGEN (e.g., prompts), the number of generated test cases, the failure rate, and reasons for failure, as well as two concrete examples, are provided in Appendix A.2.

## 3.5 HARDTESTS: 26.6K PROBLEMS WITH HIGH-QUALITY TEST CASES

We collect algorithmic coding problems and their oracle programs from five direct data sources: Codeforces, AtCoder, Luogu, CodeContests (Li et al., 2022), and TACO (Li et al., 2023). In total, these problems originate from 13 online judge platforms. The detailed statistics of these problems and their oracle programs are in Appendix A.3. We then apply HARDTESTGEN to synthesize test cases for them. After validation and filtering, we develop HARDTESTS, a large-scale dataset comprising 26.6k problems with high-quality test cases.

**Cleaning, deduplication, and decontamination.** For problems with only non-English descriptions, we translated them into English using GPT-4o. To handle overlapping content among the five direct data sources, we filtered out duplicated problems using problem IDs and n-gram overlaps in description. For correct programs, we retained all available versions and annotated them with their respective sources. We also conduct decontamination (details in Appendix A.3) by removing the problems that are in LiveCodeBench (Jain et al., 2025b) from our dataset.

**Labelling problem difficulty.** In the experiments presented in Section 4, we use the difficulty labels from Luogu, as it provides consistent and fine-grained labels for problems from both AtCoder and Codeforces. Luogu's difficulty labels are divided into seven levels, with the first level representing beginner-level problems and the seventh level corresponding to problems at the level of national Informatics Olympiad competitions.

## 4 DIRECT EVALUATION OF TEST CASE QUALITY

### 4.1 EVALUATION CRITERIA

We regard the testing of candidate programs as a binary classification process: a program is classified as positive if it passes all test cases, and negative otherwise. To directly assess the quality of test cases, we evaluate how good they are as binary classifiers. Given a problem, we categorize the candidate programs by their actual correctness (from oracle test cases or online judge platforms) and their predicted correctness (from our generated tests). When a program is both actually correct and predicted as correct, it's a true positive (TP). When a program is actually wrong but is predicted as correct, it's a false positive (FP). Similarly, we can define true negatives (TN) and false negatives

(FN). With these categories defined, we use precision and recall to measure test quality:

$$\text{Precision} = \frac{TP}{TP + FP} = \frac{\text{\# of \textbf{correct programs} that are also \textbf{predicted as correct} by tests}}{\text{\# of programs that are \textbf{predicted as correct} by the tests}},$$

$$\text{Recall} = \frac{TP}{TP + FN} = \frac{\text{\# of \textbf{correct programs} that are also \textbf{predicted as correct} by tests}}{\text{\# of \textbf{correct programs}}}.$$

Intuitively, **a higher precision implies "harder tests"** because fewer incorrect programs pass, while **a higher recall implies "more correct tests"** because fewer correct programs fail the tests.

## 4.2 Evaluation Protocol

To evaluate the accuracy of rewards that our tests can give to model training, we evaluate the precision and recall over candidate programs generated by LLMs and written by humans on subsets of problems in HARDTESTS. We compare HARDTESTS with tests from CodeContests (Li et al., 2022) and TACO (Li et al., 2023), and we also conduct ablation studies by only using a subset of the LLMGen, RPGen, and SPGen to demonstrate the necessity for all test types in HARDTESTS. More details about the evaluation protocol can be found in Appendix A.5.

To compare our tests with other synthesizers, we choose a test set of 1253 problems that exist in both HARDTESTS and the baseline datasets whenever possible. For problems from Codeforces, we select 600 problems that exist in HARDTESTS, CodeContests, and TACO. For problems from AtCoder, we select 653 problems that exist in both HARDTESTS and TACO. Because the CodeContests dataset contains very few problems originating from AtCoder and the authors did not release the code used for test case generation, we re-implemented the procedure described in their paper to construct the corresponding test cases. In total, this gives 1253 problems in the combined evaluation set. In addition, we make use of the `MatrixStudio/Codeforces-Python-Submissions` dataset, which provides a large number of human-written submissions along with their official verdicts. Since not all problems in the combined evaluation set are covered in this dataset, we randomly sample 800 Codeforces problems from it for our human-submission experiments.

We evaluate tests on candidate programs generated by LLMs and written by humans. For the 1253-problem combined evaluation set, we generate candidate programs from three LLMs: Qwen2.5-Coder-7B-Instruct (Yang et al., 2024), Qwen2.5-Coder-14B-Instruct, and GPT-4o. For each problem, we sample 10 programs from each LLM with a temperature of $0.7$ and a top-$p$ of $0.95$. For human-written programs, we rely on the 800 sampled Codeforces problems from the MatrixStudio dataset and randomly select 10 submissions per problem.

We need ground-truth labels to compute precision and recall. For AtCoder, we run candidate programs on official tests that have been previously made available. For Codeforces, LLM-generated programs are submitted to the online judge platform to obtain official verdicts, while human submissions directly come with official verdicts from the MatrixStudio dataset. We then use the synthetic test cases to classify the correctness of these programs and compare the results against the ground-truth labels, thereby evaluating test case quality.

**Ablative Baselines.** We further evaluate HARDTESTGEN under restricted test settings. In HARDTESTS, there are 4 types of test cases: LLMGen, RPGen, SPGen, and HackGen. Because RPGen and SPGen are mutually exclusive (each problem contains exactly one of them), we cannot isolate one of them in ablation. Therefore, we report two meaningful ablation settings: 1) only LLMGen, which very much resembles many existing test synthesis methods, such as KodCoder (Xu et al., 2025), as all the inputs are directly generated by LLMs, denoted as "HT–L" in Table 1, and 2) LLMGen + (RPGen or SPGen), denoted as "HT–L+R/S" in Table 1.

## 4.3 Results

Using test cases from TACO, CodeContests, and HARDTESTS, we evaluate the predicted correctness of 1) programs generated by three LLMs on the combined set of 1253 problems from AtCoder and Codeforces, and 2) programs written by human programmers on 800 Codeforces problems. By comparing the predicted correctness with the ground-truth correctness of programs, we compute the precision and recall of tests. The overall results are shown in Table 1. In Appendix A.4, we

Table 1: Precision and recall of the test cases of TACO, CodeContests, HARDTESTS, and ablative baseline on the combined dataset of problems from AtCoder and Codeforces. HT–L refers to the results using only the test cases of LLMGen from HARDTESTS. while HT–L+RS refers to the results using only the test cases of LLMGen, in addition to RPGen, or SPGen from HARDTESTS.

| | Difficulty 1 | | Difficulty 2 | | Difficulty 3 | | Difficulty 4+ | | Average | |
| --- | --- | --- | --- | --- | --- | --- | --- | --- | --- | --- |
| | prec. | recall | prec. | recall | prec. | recall | prec. | recall | prec. | recall |
| *Qwen2.5-Coder-7B-Instruct* | | | | | | | | | | |
| TACO | **96.41** | 79.5 | 75.96 | 80.92 | 53.81 | 65.47 | 17.83 | 90 | 61 | 78.97 |
| CodeContests | 92.6 | 92.67 | 63.86 | 85.69 | 39.3 | 65.57 | 10.81 | **100** | 51.64 | 85.98 |
| HT–L | 88.28 | **98.66** | 44.42 | **99.29** | 29.02 | **76.18** | 7.97 | 95 | 42.42 | **92.28** |
| HT–L+R/S | 94.97 | 98.31 | 53.18 | **99.29** | 62.8 | 75.43 | 47.73 | 95 | 64.67 | 92.01 |
| HARDTESTS | 95.17 | 98.01 | **94.95** | 98.32 | **70.83** | 75.43 | **55.88** | 95 | **79.21** | 91.69 |
| *Qwen2.5-Coder-14B-Instruct* | | | | | | | | | | |
| TACO | 92.75 | 80.8 | 86.78 | 76.64 | 66.99 | 73.6 | 34.07 | 84.52 | 70.15 | 78.89 |
| CodeContests | 90.03 | 94.55 | 76.53 | 80 | 56.35 | 85.27 | 24.14 | **98.59** | 61.76 | 89.6 |
| HT–L | 88.58 | **99.4** | 55.99 | **100** | 50.6 | **90.87** | 17.12 | **98.59** | 53.07 | **97.22** |
| HT–L+R/S | 91.49 | 98.91 | 67.42 | **100** | 74.79 | 90.21 | 59 | 95.34 | 73.18 | 96.12 |
| HARDTESTS | **93.09** | 98.91 | **91.32** | 98.34 | **82.05** | 90.21 | **59.68** | 93.93 | **81.54** | 95.35 |
| *GPT-4o* | | | | | | | | | | |
| TACO | **99.81** | 76.02 | 97 | 76.46 | 90.86 | 74.53 | 63.31 | 74.76 | 87.75 | 75.44 |
| CodeContests | 99.49 | 94.4 | 94.84 | 85.71 | 86.66 | 84.17 | 57.66 | 91.56 | 84.66 | 88.96 |
| HT–L | 99.01 | 98.54 | 94.41 | **98.93** | 82.72 | **93.43** | 47.29 | **99.82** | 80.86 | **97.68** |
| HT–L+R/S | 99.22 | **99.05** | 97 | 98.31 | 91.99 | 92.53 | 76.57 | 97.75 | 91.2 | 96.91 |
| HARDTESTS | 99.22 | 98.76 | **97.18** | 98.24 | **94.12** | 92.53 | **82.37** | 96.35 | **93.22** | 96.47 |
| *Human Submission* | | | | | | | | | | |
| TACO | **96.28** | 88.89 | **91.48** | 81.59 | **75.9** | 78.84 | 62.23 | 73.77 | **81.47** | 80.77 |
| CodeContests | 94.15 | 90.06 | 87.47 | 89.99 | 73.11 | 85.1 | 56.8 | 79.88 | 77.88 | 86.26 |
| HT–L | 83.5 | **95.57** | 69.73 | **95.97** | 54.7 | 93.59 | 42.82 | **91.72** | 62.69 | **94.21** |
| HT–L+R/S | 91.73 | 94.22 | 83.79 | 95.17 | 70.95 | **93.89** | 60.81 | 89.35 | 76.82 | 93.16 |
| HARDTESTS | 93.29 | 94.13 | 85.15 | 95.05 | 73.71 | 93.59 | **64.16** | 89.35 | 79.08 | 93.03 |

also report results separately for the AtCoder subset and the Codeforces subset of the combined evaluation set. We present qualitative analyses of synthetic tests in Appendix A.6.

We find that **HARDTESTS significantly outperforms TACO and CodeContests in terms of both precision and recall under most evaluation settings.** Moreover, this advantage becomes more pronounced as problem difficulty increases. For example, for the Qwen2.5-Coder-7B-Instruct model on problems with difficulty level 4+, TACO achieves a precision of 17.83, whereas HARDTESTS achieves a precision of 55.88, more than 3x that of TACO. This implies that using HARDTESTS during RL training would yield more true positive rewards and fewer false positive rewards. Furthermore, we observe the precision advantage of HARDTESTS becomes more pronounced as the source of programs becomes less "intelligent" (ranging from human-written to 7B LLM-generated). We attribute this to the fact that less skilled programmers are more likely to produce functionally correct but inefficient programs. For instance, among incorrect human-written programs, 14.9% are due to TLE (Time Limit Exceeded), whereas among the incorrect programs written by the three LLMs, 30.0% are due to TLE. Consequently, the larger and more diverse test cases in HARDTESTS are more likely to catch inefficient programs than the small-scale test cases in TACO and CodeContests.

Compared with the ablative baselines in Table 1, HARDTESTS that includes RPGen, SPGen, and HackGen almost consistently leads to a precision improvement ranging from 0.2% to 40%, while the decrease in recall is always within 1%. This demonstrates the necessity for having all types of tests.

For Table 1, we use GPT-4o to generate test cases, but we also discuss the use of other LLMs for test case generation in Appendix A.9. Our results suggest that HARDTESTGEN can also perform well with recent open-weight LLMs, demonstrating its generalizability.

Table 2: pass@k (%) LLMs after rejection sampling based on Qwen3-4B on LiveCodeBench-105.

|  | pass@1 | pass@10 |
|---|---|---|
| Qwen3-4B | **38.48** | 56.19 |
| Qwen3-4B (with *bad 5k*) | 34.00 | 54.92 |
| Qwen3-4B (with *random 5k*) | 32.75 | 57.14 |
| Qwen3-4B (with *good 5k*) | 36.00 | **60.00** |

Table 3: pass@k (%) for LLMs RL-trained from Qwen3-4B on LiveCodeBench-105.

|  | pass@1 | pass@10 |
|---|---|---|
| Qwen3-4B | 38.48 | 56.19 |
| Qwen3-4B-TACO | 36.95 | 57.14 |
| Qwen3-4B-HT | **39.42** | **64.76** |

## 5 DOWNSTREAM EFFECTS OF TEST CASE QUALITY IN LLM POST-TRAINING

In this section, we aim to answer two questions with HARDTESTS: 1) when does verifier/test quality matter, and 2) how much does it matter in post-training? We run experiments in two different post-training scenarios: *rejection sampling*, and *reinforcement learning*. We present the results below and show that verifier quality impacts these two scenarios significantly.

### 5.1 EXPERIMENT SETUP

**Rejection sampling.** Fine-tuning a model with its own reasoning trajectories can also improve its reasoning ability (Zelikman et al., 2022). Hence, determining which trajectories to use is a critical issue. To examine the effects of test quality, we sampled 5 traces of Qwen3-4B and used the tests generated by HARDTESTGEN for filtering. We selected 4989 questions where there is at least one Qwen3-4B generated program that passes the tests and at least one that fails the tests. We create 3 datasets for rejection sampling, each containing one trajectory per question. The *bad 5k* randomly samples one incorrect trajectory for each question. The *good 5k* randomly samples one correct trajectory. The *random 5k* randomly samples one trajectory, regardless of its correctness, for each question. We further fine-tune Qwen3-4B with these 3 datasets and compare the performance of the resulting models. All our fine-tuning experiments were done with Llama-factory (Zheng et al., 2024).

**Reinforcement learning.** Verifier feedback is an option for distillation, but it is a must for reinforcement learning. To investigate how verifier quality affects RL, we train Qwen3-4B with RL using the same problem set, the identical training setup, and different test cases. We select a problem set with ∼5k problems that exist in both HARDTESTS and TACO for training. We use a modification of veRL (Sheng et al., 2024) inspired by Code-R1 (Liu & Zhang, 2025) for training with GRPO (Shao et al., 2024). When a program passes all tests, it gets a reward of 1, otherwise, it gets a reward of 0. We compare different verifiers by looking at the final performance and the validation curve.

**Evaluation protocol.** We use LiveCodeBench (Jain et al., 2025b) version 5 to evaluate the model performance. Since all the programs we use for tuning are in C++, we build an evaluation pipeline for evaluating C++ programs for LiveCodeBench and select a 105-problem subset where all problems require reading from and writing to standard I/O. We name this subset "LiveCodeBench-105". Details about our training and evaluation procedure can be found in Appendix A.7, including the problems and hyperparameters we use for training and the sampling parameters we use for evaluation.

### 5.2 RESULTS

**Rejection sampling performance is highly dependent on sample quality and needs a good verifier.** We evaluated variants of Qwen3-4B models trained with rejection sampling from different 5k subsets on LiveCodeBench-105 and present the results in Table 2. Model trained from incorrect samples identified by HARDTESTGEN's tests drops more significantly in pass@k. Rejection sampling with randomly selected data could harm pass@1 even more, despite the slight improvements in pass@10. In contrast, using a 5k subset verified by HARDTESTGEN's test cases results in a smaller drop in pass@1 and a notable gain in pass@5 and pass@10, suggesting that verifiers are important to rejection sampling.

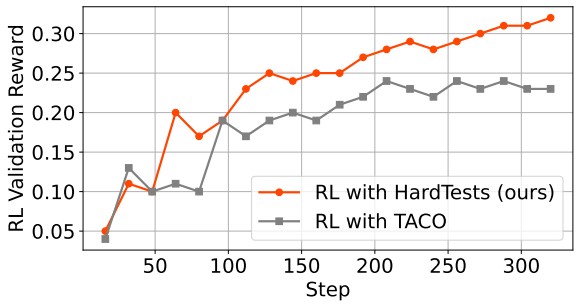

Figure 3: RL Validation Rewards Over Time. Reward from HARDTESTS makes the training better. overall performance.

**Test quality matters significantly for reinforcement learning.** As shown in Figure 3, the validation reward curve for HARDTESTS during RL training is generally higher than that for TACO. This indicates that for the same problems, HARDTESTS is giving better rewards. To evaluate on LiveCodeBench-105, we run the best checkpoints (according to valid reward) of both training jobs within 100 steps. As reported in Table 3, TACO tests hurt the model's overall performance, while HARDTESTS improves the model's

## 6 CONCLUSION AND FUTURE WORK

We present HARDTESTGEN, an LLM-based test synthesis pipeline, which is used to create HARDTESTS, an algorithmic coding dataset with 26.6k problems and significantly higher-quality tests. We examine when and how much test quality matters in LLM post-training, showing that harder tests generated by HARDTESTGEN can indeed help LLM post-training in many scenarios. While HARDTESTGEN assumes the existence of oracle solutions, we briefly discuss an initial idea for synthesizing tests without oracles in Appendix A.8. We envision two future directions: 1) to develop better methods for synthesizing tests without an oracle, and 2) to apply HARDTESTGEN to both stateless and stateful real-world coding problems. It is worth noting that stateful computations can often be transformed into equivalent stateless representations using design patterns such as monads (Wadler, 1995).

## 7 ACKNOWLEDGEMENT

This project is partially supported by a research gift from Amazon AGI and a gift from SCOP Venture to CMU Li Lab. The OpenAI API credits used in this paper were partially supported by the OpenAI Research Access Program. The training compute used was partially supported by National Center for Supercomputing Applications. KZ was partially supported by ChipAgents.ai. We thank Luogu for granting permission to crawl and use their data as part of the HARDTESTS dataset.

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

# A    APPENDIX

## A.1    MORE RELATED WORK DISCCUSION

**Early approches.**    Early approaches in test generation employ search-based heuristics methods (Fraser & Arcuri, 2011; Lukasczyk & Fraser, 2022), or fuzzing to expose software vulnerabilities (Fioraldi et al., 2020). These methods often yield high code coverage, while high code coverage is likely to improve fault detection, it is not guaranteed (Cai & Lyu, 2005).

**LLM-based approches in software testing.** Some LLM-based approaches have been introduced to improve coverage (Ryan et al., 2024; Wang et al., 2024; Altmayer Pizzorno & Berger, 2025) or uncover bugs in plausible programs using differential testing (Liu et al., 2024), respectively. However, the target of these methods is different from using testing as a reward signal for RL.For instance, there is not a single program under test or focal method. RL involves testing hundreds of programs per problem, making coverage-based methods less tractable.

**Synthetic test quality and its implications.** Although existing LLM test synthesis methods prove to be useful in many scenarios, such as improving the quality of synthetic data (Wei et al., 2024) and software engineering(Mündler et al., 2025; Jain et al., 2024), their quality is far from perfect (Yuan et al., 2024) and are bounded in complexity, because direct generations of complicated data structures often result in inconsistency (Zhang et al., 2023). Weak verifiers can harm downstream code generation and search performance (Light et al., 2025).ǎ The quality of those synthetic tests and their implications are less discussed. Existing benchmarks for LLM test case generation abilities focus on code coverage and/or mutation scores (Wang et al., 2025a; Zhang et al., 2024; Jain et al., 2025a; 2024), the success rate for reproducing issues (Mündler et al., 2025), and the code change coverage for generated code patches (Ahmed et al., 2024; Mündler et al., 2025).

## A.2    DETAILS OF THE TEST CASES GENERATION PIPELINE OF HARDTESTGEN

In this section, we first introduce the prompts used in HARDTESTGEN in Appendix A.2.1. We will then provide the statistics of the test cases in HARDTESTS in Appendix A.2.2 and discuss the failure rate and the major failure reasons of HARDTESTGEN in Appendix A.2.3. Finally, we will include two examples of HARDTESTGEN in Appendix A.2.4.

### A.2.1    PROMPTS USED IN HARDTESTGEN

**Prompt for the generation of the input validator and output judging function.**    We use the following LLM prompt to generate an input validator function, and an output judge function when necessary. This prompt includes the problem specification and the oracle program to help the LLM have a better understanding.

```
1  I have a competitive programming problem. To test the correctness of candidate programs,
   ↪  I need to create many test cases.
2
3  Each test case is an input-output pair. The input part will be fully provided as stdin to
   ↪  the candidate program, and then the candidate output will be collected from stdout.
   ↪  In most cases, we determine the correctness of the program by comparing the candidate
   ↪  output with the output part of the test case (i.e., the reference output), while
   ↪  sometimes, we need to use a custom function to judge the correctness of the candidate
   ↪  output, instead.
4
5  Note: Sometimes, a problem may require a single test case to contain multiple sub-tasks.
   ↪  For example: the first line of the input contains an integer $t\ (1 \leq t \leq
   ↪  1000)$, followed by inputs of $t$ independent sub-tasks. The problem statement may
   ↪  sometimes refer to a sub-task as a "test case", but this is merely a difference in
   ↪  terminology.
6
7  # Input Validating Function
8
```

9   Suppose I have already written some input generator functions, and used them to generate
    ↪   many test case inputs. However, since they are randomly generated, they may not fully
    ↪   adhere to the constraints specified in the problem. I need you to filter out invalid
    ↪   test cases.

10

11  Given the problem described below, please identify value type and range, and list all
    ↪   constraints input data. Please generate a Python function `validate_input(input_str:
    ↪   str) -> bool` that checks the input against all constraints and returns a boolean
    ↪   indicating whether the input is valid according to the problem statement.

12

13  However, if a constraint cannot be verified within a reasonable time complexity (e.g.,
    ↪   $O(n)$ for $n \leq 10^6$, or $O(n^2)$ for $n \leq 10^3$), or if it makes the code too
    ↪   complex, then it can be skipped.

14

15  **Pay close attention**: If the problem says "It's guaranteed that...", then what follows
    ↪   is precisely something that must be verified. This is because the so-called
    ↪   "guarantee" in the problem is typically enforced through the input validating
    ↪   function, so you must validate it in `validate_input`. Of course, only if it can be
    ↪   done in reasonable time complexity.

16

17  **Example 1**: Cicasso has $n$ sticks of lengths $l = (l\_0, l\_1, \dots, l\_{n-1})$. But
    ↪   these $n$ sticks cannot form a convex polygon with non-zero area. You need to add one
    ↪   stick so that the resulting $n+1$ sticks can form such a polygon. The input consists
    ↪   of two lines: the first line is an integer $n$ ($3 \leq n \leq 10^5$). The second
    ↪   line has $n$ integers $l\_i$ ($1 \leq l\_i \leq 10^9$).

18

19  The `validate_input` function should not only check that $n$ and $l\_i$ are within the
    ↪   correct range and that there are exactly $n$ numbers in the second line, but also
    ↪   check that the $n$ sticks cannot form a convex polygon with non-zero area, i.e., that
    ↪   the longest stick is greater than or equal to the sum of the rest.

20

21  **Example 2**: Suppose there is a permutation $p = (p\_0, p\_1, \dots, p\_{n-1})$ of numbers
    ↪   from 1 to $n$ ($1 \leq n \leq 2 \times 10^5$). But you do not know the permutation
    ↪   $p$. Instead, you are given an array $s = (s\_0, s\_1, \dots, s\_{n-1})$, where $s\_i$ is
    ↪   the sum of all $p\_j < p\_i$ for $j < i$. Your task is to recover $p\_i$.

22

23  In theory, we should verify whether the $s\_i$ values correspond to a valid permutation
    ↪   $p\_i$, but that requires solving for $p\_i$, which is too complex. Moreover, when
    ↪   generating inputs, it's quite easy to ensure that the $s\_i$ comes from a valid
    ↪   permutation, so mistakes are unlikely. (Note: If verifying a constraint isn't too
    ↪   complex, you should still check it.) Therefore, we only need to check that $n$ is
    ↪   within the range and that $s$ has exactly $n$ elements.

24

25  # Output Judging Function

26

27  Given the problem described below, please generate an output judging function in Python
    ↪   to compare the equivalence of two output results. The function takes an input string
    ↪   (can actually represent a list of numbers or strings) and two output strings as
    ↪   arguments, and returns a boolean value. For example, you should use set comparison
    ↪   when the order of results does not matter.

28

29  In most cases, we can determine whether the two output strings (`candidate_output` and
    ↪   `reference_output`) are equivalent using string comparison. The specific function is
    ↪   shown below.

30

31  ```python
32  def output_judging_function(input_str: str, candidate_output: str, reference_output:
    ↪   str) -> bool:
33  ^^Inormalized_candidate_output = '\n'.join(line.rstrip() for line in
    ↪   candidate_output.rstrip().splitlines())
34      normalized_reference_output = '\n'.join(line.rstrip() for line in
        ↪   reference_output.rstrip().splitlines())
35      return normalized_candidate_output == normalized_reference_output
36  ```

37

38 However, for a few problems, the above `output_judging_function` does not work.

39

40 **Example 1**: The problem asks to output a list (`List[int]`), but the order of elements
   ↪ in the list does not matter.

41

42 In this case, we should convert both `candidate_output: str` and `reference_output: str`
   ↪ into `List[int]`, sort them, and then compare them.

43

44 **Example 2**: Given a graph with both directed and undirected edges, you must make all
   ↪ undirected edges directed so that the resulting graph has no cycles. If it is
   ↪ possible, output "YES" and the resulting graph (list of directed edges), otherwise
   ↪ output "NO".

45

46 Here, in `output_judging_function`, we should first determine from `reference_output`
   ↪ whether a solution is possible. If both `candidate_output` and `reference_output`
   ↪ say "YES", then we should also validate whether the graph provided in
   ↪ `candidate_output` is valid: check whether all edges exist in the input and whether
   ↪ the graph is acyclic (e.g., via DFS).

47

48 **Example 3**: There are a total of $T$ sub-tasks. Each sub-task gives a pair of integers
   ↪ $l, r$ ($1 \leq l \leq r \leq 998244353$), and the goal is to find a pair of integers
   ↪ $x, y$ such that $l \leq x, y \leq r$, $x \ne y$, and $y$ is divisible by $x$. It is
   ↪ guaranteed that every sub-task has a valid solution.

49

50 For each pair $x, y$ provided in the `candidate_output`, simply check whether they
   ↪ satisfy all the conditions mentioned in the problem statement. The
   ↪ `output_judging_function` for this problem does not need to use the
   ↪ `reference_output`; it only requires the `input_str`.

51

52 You need to first analyze whether this particular problem requires a custom
   ↪ `output_judging_function` (different from the one given above). If yes, generate a
   ↪ custom `output_judging_function`. If not, don't output it. Sometimes only
   ↪ `input_str` is needed and `reference_output` is not required; other times only
   ↪ `reference_output` is needed and `input_str` is not required; and in some cases,
   ↪ both are needed. However, regardless of which ones are actually used, the function
   ↪ signature must always be: `output_judging_function(input_str: str, candidate_output:
   ↪ str, reference_output: str) -> bool`.

53

54 Generally speaking, if a problem states "there are multiple possible answers, any one is
   ↪ acceptable," this implies that the problem requires a custom output judging function.
   ↪ However, even if this is not explicitly mentioned, the problem may still actually
   ↪ require a custom output judging function. You need to determine this yourself.

55

56 ---

57

58 Also, when generating the above two functions, some known tricks or conclusions may be
   ↪ helpful, and you should derive them yourself if needed. I will give you the correct
   ↪ solution to the problem, and you can use it to derive certain conclusions or tricks.

59

60 Your output format must strictly follow:

61

62 # Analysis

63

64 ... (Analyze the problem, constraints, how to generate the input validating function and
   ↪ output judging function, etc.)

65

66 # Result

67

68 ```json

69 {

70 ^^I"input_validating_function": "A block of Python code containing the `validate_input`
   ↪ function. No other content.",

71 ^^I"needs_custom_output_judging_function": true or false,

72 ^^I"output_judging_function": "A block of Python code containing the
   ↪ `output_judging_function` function. No other content." or null

```
73   }
74   ```
75
76   ---
77
78
79   Note:
80   * All your code should be in Python 3.
81   * Do not wrap the Python code in ```python```, just provide it plainly.
82   * The Python code block under each field should be independent. In other words, they
     ↪   should not call or reference each other. If one block imports a library, other blocks
     ↪   must re-import it as needed.
83   * In a Python block, you should first import the necessary libraries, and then start
     ↪   defining functions. Important: Do not place import statements inside the functions.
84   * Only Python's built-in libraries are permitted for import.
85
86   For example, a block of Python code for input validating function should look like this:
87
88   import ... (some modules)
89
90   def validate_input(input_str: str) -> bool:
91   ^^I... (some code)
92
93   A block of Python code for output judging function (if needed) should look like this:
94
95   import ... (some modules)
96
97   def output_judging_function(input_str: str, candidate_output: str, reference_output:
     ↪   str) -> bool:
98   ^^I... (some code)
99
100  ---
101
102  # Problem Statement
103
104  {{ problem_specification }}
105
106  ---
107
108  # Correct Program
109
110  {{ oracle_program }}
111
```

**Prompt for the generation of input-generating functions.** We use the following prompt to have the LLM directly generate small-scale test inputs (LLMGen), and functions that can produce test inputs (RPGen, SPGen, and HackGen). This prompt makes use of the problem specification, oracle program, and input validator to help the LLM better understand the problem requirements.

```
1   I have a competitive programming problem. To test candidate programs' correctness, I need
    ↪   to create many test cases.
2
3   Each test case is an input-output pair. The input part will be fully provided as stdin to
    ↪   the candidate program, and then the candidate output will be collected from stdout.
    ↪   In most cases, we determine the correctness of the program by comparing the candidate
    ↪   output with the output part of the test case (i.e., the reference output), while
    ↪   sometimes, we need to use a custom function to judge the correctness of the candidate
    ↪   output, instead.
4
5   Note: Sometimes, a problem may require a single test case to contain multiple sub-tasks.
    ↪   For example: the first line of the input contains an integer $t\ (1 \leq t \leq
    ↪   1000)$, followed by inputs of $t$ independent sub-tasks. The problem statement may
    ↪   sometimes refer to a sub-task as a "test case", but this is merely a difference in
    ↪   terminology.
```

```
6
7  Since the output part can be obtained by running correct programs, I only need you to
   ↪    help me generate the input part.
8
9  The input should comply with the constraints given in the problem statement. I will give
   ↪    you an input validating function that checks whether the input meets all the
   ↪    constraints specified in the problem statement. However, some constraints may not be
   ↪    checked by the input validating function due to the difficulty of verification.
   ↪    Nevertheless, the input you generate should still comply with all of these
   ↪    constraints.
10
11 There are four types of input: LLMGen, RPGen, SPGen, and HackGen. But note that RPGen and
   ↪    SPGen are mutually exclusive.
12
13 # LLMGen
14
15 Please generate {{ num_LLMGen_input }} valid inputs according to the problem stated
   ↪    below. You may follow the examples given in the problem statement and generate
   ↪    variations that are different. Please respect the constraints and data types in the
   ↪    problem statement.
16
17 Note: each input's length should be similar to the sample test cases' input, comply with
   ↪    the constraints given in the problem, and must not exceed 300 characters under any
   ↪    circumstances. If it is not possible to generate input under this length limit, give
   ↪    up on generating them.
18
19 # RPGen
20
21 Given the problem described below, please identify the input data types (e.g., int,
   ↪    string), ranges, and data constraints (e.g., x-y coordinates forming a convex
   ↪    polygon or number of items which should be nonnegative), and then generate a python
   ↪    function `gen_RPGen_input` which will return one random input data for the problem
   ↪    with respecting to these types, ranges, and constraints.
22
23 You should ensure the generated input satisfies the constraints as much as possible, and
   ↪    may even sacrifice some degree of randomness to do so. But if trying to enforce a
   ↪    constraint leads to a function that cannot run within finite and reasonable time
   ↪    complexity (e.g., $O(n)$ for $n \leq 10^6$, or $O(n^2)$ for $n \leq 10^3$), then you
   ↪    may ignore that constraint.
24
25 **Pay close attention**: do not use `while` loops, especially ones that "keep generating
   ↪    until a constraint is satisfied." That can cause unlimited running time and make
   ↪    input generation fail.
26
27 Some problems may require certain test cases to satisfy specific constraints (for
   ↪    example, 10% of test cases satisfy $n \leq 100$, 10% of the test cases satisfy $n
   ↪    \leq 1000$, etc.). Ignore this requirement. All test cases should be generated
   ↪    according to the most general constraints.
28
29 Sometimes, generating input that satisfies the constraints requires some trick. You need
   ↪    to deduce it yourself (e.g., the example below about when $n$ sticks cannot form a
   ↪    convex polygon). I will give you the correct solution for the problem, and you can
   ↪    analyze it to discover some tricks or conclusions.
30
31 **Example 1**: Cicasso has $n$ sticks ($3 \leq n \leq 10^5$) of lengths $l_i$ ($1 \leq l_i
   ↪    \leq 10^9$, for $i=0,1,\dots,n-1$). But these $n$ sticks cannot form a convex polygon
   ↪    of non-zero area. You need to add one more stick, so that the $n+1$ sticks can form a
   ↪    convex polygon of non-zero area. Output the minimum length of the additional stick.
32
```

33  We can randomly generate $n \in [3, 10^5]$, but cannot randomly generate $l_i$, because
    ↪  such $l_i$ will likely not satisfy the constraint that the $n$ sticks cannot form a
    ↪  convex polygon of non-zero area. (It's not feasible to randomly generate and then
    ↪  filter, since it's too time-consuming.) We know that this constraint actually
    ↪  requires "the maximum $l_i$ is greater than or equal to the sum of all the other
    ↪  $l_i$." So we can first randomly sample a $l_0$ in $[n-1, 10^9]$ as the maximum
    ↪  $l_i$, then sample an integer $s \in [n-1, l_0]$ as the total sum of the other $l_1,
    ↪  \dots, l_{n-1}$, and finally use a partitioning trick to sample $l_1, \dots, l_{n-1}$
    ↪  such that each element is at least 1 and the total sum is $s$. After that, we can
    ↪  shuffle the $l_i$ list.

34
35  **Example 2**: There is a permutation $p = (p_0, p_1, ..., p_{n-1})$ of numbers from 1 to
    ↪  $n$ ($1 \leq n \leq 2\cdot 10^5$). You do not know this permutation, but you are
    ↪  given an array $s = (s_0,\dots,s_{n-1})$, where $s_i$ is the sum of all $p_j < p_i$
    ↪  with $j < i$. Find $p_i$.

36
37  We can first randomly generate $n \in [1, 2 \times 10^5]$. But we cannot directly
    ↪  generate an array $s_i$ randomly, because it is very unlikely to satisfy the
    ↪  constraints. Instead, we should reverse the process: first generate a random
    ↪  permutation $p_i$, and then compute the corresponding $s_i$.

38
39  **Example 3**: This problem has $t \in [1, 1000]$ groups of independent sub-tasks. Each
    ↪  sub-task has an integer $n \in [1, 10^5]$ and an array $a$ of length $n$, where $a_i
    ↪  \in [1,10^5]$. The problem guarantees that the total sum of all $n$ across all $t$
    ↪  sub-tasks does not exceed $2 \times 10^5$.

40
41  We can first randomly generate $t \in [1, 1000]$. But at this point we cannot directly
    ↪  sample $t$ values of $n$ from $[1, 10^5]$, because their sum is likely to exceed $2
    ↪  \times 10^5$. So instead, we randomly sample $s \in [t, 2\times 10^5]$, and then
    ↪  partition $s$ into $n_0, n_1, \dots, n_{t-1}$ such that each value is at least 1 and
    ↪  their sum is $s$.

42
43  The following Python function demonstrates how, given positive integers $m$ and $s$, with
    ↪  $m \leq s$, one can randomly select $m$ positive integers such that their sum equals
    ↪  $s$. This is just for your reference.

44
45  ```python
46  import random
47
48  assert m <= s
49  if m >= 2:
50  ^^Ibreaks = random.sample(range(1, s), m - 1)
51  ^^Ibreaks.sort()
52  ^^Iresults = [breaks[0]] + [breaks[i] - breaks[i - 1] for i in range(1, len(breaks))] +
    ↪  [s - breaks[-1]]
53  else:
54  ^^Iresults = [s]
55  ```
56
57  # SPGen
58
59  For most problems, there is only one type of output. But there are some problems where
    ↪  outputs fall into multiple categories. These are called multi-category output
    ↪  problems. For example, some problems require the output to be "Yes" or "No", while
    ↪  others ask you to output the solution if it exists, otherwise output -1. In such
    ↪  cases, if we treat it as a regular problem and only write a single
    ↪  `gen_range_based_input` function to generate inputs randomly, the resulting outputs
    ↪  will be very imbalanced. For example, the "Yes" outputs may require special
    ↪  construction, so nearly all generated inputs produce "No" as the answer. Thus, even a
    ↪  candidate program that always prints "No" would pass all test cases. Therefore, for
    ↪  such problems, we do not generate `gen_RPGen_input` function.
60

```
61   Given the problem described below, please identify how many categories are there in the
   ↪    output values, denote as $m_S$. For each categoy, please generate one function
   ↪    `gen_SPGen_input_<category_label>` in python which will produce a random input data
   ↪    that will output one value in the given category. Please replace `<category_label>`
   ↪    with the inferred output value categories.
62
63   Each time the function is called, it should be able to generate--within reasonable time
   ↪    complexity--one random input that satisfies the constraints and whose corresponding
   ↪    output belongs to the corresponding category. If it is difficult to write a function
   ↪    that randomly generates some category, you can:
64
65   1. Sacrifice randomness and perform special construction, even returning a fixed value
   ↪    each time
66
67   or 2. Construct completely random data, similar to `gen_RPGen_input`
68
69   Sometimes, a problem may require a single test case to contain multiple independent
   ↪    sub-tasks. In this case, each sub-task in each input generated by
   ↪    `gen_SPGen_input_<category_label>` should have the corresponding output category,
   ↪    e.g., all corresponding outputs should be "No".
70
71   **Example 1**: Given two $n \times m$ binary matrices $A, B$. You can take the following
   ↪    operation: select a rectangle in matrix $A$ with height and width both at least 2,
   ↪    and flip the values at the four corner positions. You are to answer whether it's
   ↪    possible to make $A$ equal to $B$ using this operation. If possible, output "Yes" and
   ↪    the resulting matrix; otherwise, output "No".
72
73   There are two outputs here: "Yes" and "No", corresponding to two categories of inputs.
   ↪    For the first category, we create `gen_SPGen_input_yes`, such that $A$ can be
   ↪    transformed into $B$. We can randomly construct matrix $A$, then perform $t$
   ↪    operations (you can decide $t$ yourself, but it should not be too small or too large
   ↪    to avoid long generation time), where each operation selects a rectangle and flips
   ↪    the corners. Then the result becomes matrix $B$. For the second category, we write
   ↪    `gen_SPGen_input_no`, where $A$ cannot be transformed into $B$. One way is to
   ↪    randomly flip a position in matrix $B$ from the previous construction, which makes it
   ↪    impossible. This sacrifices randomness, but is simple and acceptable.
74
75   **Example 2**: Given two numbers $n, m$ ($1\leq n \leq m\leq 5\times10^8$), you are to
   ↪    determine whether it is possible to transform $n$ into $m$ by multiplying by 2 and 3,
   ↪    and if so, output the minimum number of operations. Otherwise, output -1.
76
77   There are two outputs: the minimum operation count, and -1. Correspondingly, we have two
   ↪    input generators. For the first case, where $n$ can be transformed into $m$, we can
   ↪    randomly generate $n\in [1, 5\times 10^8]$, then perform $t$ operations (multiply by
   ↪    2 or 3) until $t$ steps are complete or further multiplication would exceed
   ↪    $5\times10^8$. The result becomes $m$. For the second case, where $n$ cannot be
   ↪    transformed into $m$, we can firstly randomly generate $m > n$, and then if $n$ can
   ↪    be transformed into $m$, simply set $m = m-1$.
78
79   **Example 3**: Player A and B are playing tic-tac-toe. Player A goes first. You are given
   ↪    a $3 \times 3$ board, where each cell is ".", "X", or "0". Output the current state,
   ↪    one of: "first" (next move is A), "second" (next is B), "illegal" (not possible in a
   ↪    legal game), "the first player won", "the second player won", or "draw".
80
81   There are 6 output categories, corresponding to 6 input categories. For the first output
   ↪    category, we need to create `gen_SPGen_input_first` where the next move is A's. We
   ↪    can randomly select $t\in[0, 4]$, then randomly place $t$ X's and $t$ 0's. This may
   ↪    lead to a win or illegal state, but we should NOT filter those during generation,
   ↪    because doing so would make the code too complex and slow. We only need most of the
   ↪    generated inputs to match this category. For the second category, place $t+1$ X's and
   ↪    $t$ 0's ($t\in [1,3]$). For the third category, it must be illegal, e.g., X and 0
   ↪    count difference is too large, or both players have already won. We can create
   ↪    `gen_SPGen_input_illegal_mark_num` and `gen_SPGen_input_illegal_both_win`, etc. Do
   ↪    the same for the remaining categories.
82
```

83  # HackGen

85  Although RPGen/SPGen inputs can guarantee large data size, for some problems,
    ↪   large-scale random data is not enough. We also need hacking inputs, which refers to
    ↪   inputs that are very tricky for candidate programs.

88  Given the problem described below, please first describe several flawed solution
    ↪   programs using brute-force enumeration or a classic algorithm (e.g., depth-first
    ↪   search), by explicitly ignoring some constraints, corner conditions, or data range.
    ↪   Secondly, please think about scenarios where these programs will fail or exceed the
    ↪   time limit (TLE). Thirdly, for each scenario, generate one function
    ↪   `gen_HackGen_input_<pattern_name>` which will produce possibly random input data
    ↪   corresponding to such a failing scenario.

90  Note: for some problems, even though the brute-force algorithm's worst-case complexity
    ↪   is $O(n^2)$, due to rare worst-case inputs, the actual runtime is closer to $O(n)$.
    ↪   In these cases, you need to specially construct the data to repeatedly trigger the
    ↪   worst-case scenario for those brute-force algorithms.

92  For some problems, we also need some types of inputs to expose bugs caused by failure to
    ↪   handle edge cases. Thus, you should think about whether there are any special edge
    ↪   cases (e.g., input $n=0$, or tree root is None, etc.). Note that the randomness of
    ↪   the input data itself at this time is not important. The key point is to expose the
    ↪   errors of the candidate programs.

94  Of course, if the problem doesn't require any HackGen input, then do not generate them.
    ↪   Especially if a HackGen input is simply large-scale data, then you shouldn't bother.
    ↪   HackGen input must be specially constructed--RPGen/SPGen input should almost never
    ↪   produce them.

96  **Example 1**: Given two numbers $n$ and $m$ ($1 \leq n \leq m \leq 5 \times 10^8$), the
    ↪   task is to determine whether it is possible to transform $n$ into $m$ by repeatedly
    ↪   multiplying $n$ by 2 or by 3. If possible, output the minimum number of operations
    ↪   required; otherwise, output -1.

98  A brute-force approach that a candidate program might take is to use DFS, recursively
    ↪   trying to multiply $n$ by 2 or 3 until it becomes greater than or equal to $m$. If we
    ↪   randomly choose $n$ and $m$, the ratio between them is usually small, so this
    ↪   approach might still pass. One kind of effective HackGen input is to set $n \in [1,
    ↪   5]$ and $m \in [4 \times 10^8, 5 \times 10^8]$. This creates a large gap between $n$
    ↪   and $m$, making the brute-force DFS approach inefficient. We can name the
    ↪   corresponding function `gen_HackGen_input_small_n_big_m`. You should consider other
    ↪   types of HIs yourself.

100 **Example 2**: Given a string $S$ of length $n \in [1, 10^5]$, we repeatedly perform the
    ↪   following operation: find two identical adjacent characters and delete them. This
    ↪   continues until there are no more identical adjacent characters in $S$.

102 This problem should be solved using a stack to achieve an $O(n)$ time complexity.
    ↪   However, some candidate programs might use a brute-force simulation approach --
    ↪   repeatedly scanning the string and removing adjacent equal characters -- which can
    ↪   result in a worst-case time complexity of $O(n^2)$. If we generate $S$ completely at
    ↪   random, it's likely that there will only be a few pairs of identical adjacent
    ↪   characters. One kind of HackGen input is to construct a string $S$ of a long even
    ↪   length (e.g., in $[5 \times 10^4, 10^5]$) and set `S[2*k] == S[2*k+1]`, thereby
    ↪   introducing a large number of adjacent equal character pairs. However, if the
    ↪   candidate program deletes all adjacent equal pairs in each round, the time complexity
    ↪   remains $O(n)$. Another HackGen input is to construct a string $S$ of a long even
    ↪   length (e.g., in $[5 \times 10^4, 10^5]$) such that `S[:n//2] == S[n//2:][::-1]`,
    ↪   which forces the program to go through $n$ rounds to completely remove all
    ↪   characters, resulting in the true worst-case time complexity of $O(n^2)$. These two
    ↪   functions can be named `gen_HackGen_input_pairwise_equal` and
    ↪   `gen_HackGen_input_mirrored_halves`, respectively.

103

```
104  **Example 3**: Given integer $w\in[1, 100]$, determine whether it can be written as the
     ↪   sum of two positive even integers.
105
106  Candidate programs may output "Yes" when $w$ is even, and "No" when $w$ is odd. But a
     ↪   special case is $w=2$, which should be "No". So we can create
     ↪   `gen_HackGen_input_two`, which always returns the string `"2"`.
107
108  Important: if a type of HackGen input is just setting data to their largest scale, then
     ↪   it is unnecessary.
109
110  ---
111
112  Your output format must strictly be as follows:
113
114  # Analysis
115
116  ...
117  (generally, you should first analyze the problem and data constraints, and then analyze
     ↪   how to generate LLMGen Input, how to generate RPGen Input, and whether the problem is
     ↪   a multi-category output problem (In that case, generate SPGen generation functions
     ↪   for each output category. Make sure you mentioned the corresponding function names in
     ↪   the Analysis part). Then you should list some naive candidate programs and analyze
     ↪   how to generate HackGen Input.)
118
119  # Result
120
121  ```json
122  {
123  ^^I"LLMGen_inputs": ["LLMGen_input_1", "LLMGen_input_2", ...],
124  ^^I"is_multi_category_output_problem": true or false,
125  ^^I"RPGen_SPGen_input_generator": "a block of Python code containing a function
     ↪   `gen_RPGen_input` (for regular problems), or multiple functions
     ↪   `gen_SPGen_input_<category_label>` (for multi-category output problems)",
126  ^^I"HackGen_input_generator": "a block of Python code containing one or more
     ↪   `gen_HackGen_input_<pattern_name>` functions" or null (if no HackGen input is needed)
127  }
128  ```
129
130  ---
131
132  Note:
133  * All your code should be in Python 3.
134  * Do not wrap the Python code in ```python```, just provide it plainly.
135  * The Python code block under each field should be independent. In other words, they
     ↪   should not call or reference each other. If one block imports a library, other blocks
     ↪   must re-import it as needed.
136  * In a Python block, you should first import the necessary libraries, and then start
     ↪   defining functions. Important: Do not place import statements inside the functions.
137  * Only Python's built-in libraries are permitted for import.
138
139  For example, a block of Python code for RPGen inputs of regular problems should look like
     ↪   this:
140
141  import ... (some modules)
142
143  def gen_RPGen_input(input_str: str) -> bool:
144  ^^I... (some code)
145
146  A block of Python code for SPGen inputs of multi-category output problem may look like
     ↪   this:
147
148  import ... (some modules)
149
150  def gen_SPGen_input_<category_label>(input_str: str) -> bool:
151  ^^I... (some code)
```

Table 4: Distribution of problems by number of test cases.

| Number of Test Cases | 1–9 | 10–19 | 20–39 | 40–80 |
|---|---|---|---|---|
| Percentage | 1.7% | 5.3% | 51.8% | 41.2% |

```
152
153  def gen_SPGen_input_<category_label>(input_str: str) -> bool:
154  ^^I... (some code)
155
156  ...
157
158  And the HackGen input block is similar.
159
160  ---
161
162  # Problem Statement
163
164  {{ problem_specification }}
165
166  ---
167
168  # Correct Program
169
170  {{ oracle_program }}
171
172  ---
173
174  # Input Validating Function
175
176  {{ input_validating_function }}
177
```

Note that in the prompts above, we provide two to three carefully crafted examples for each function that we ask the LLM to generate, enabling in-context learning. Additionally, we prompt the LLM to perform chain-of-thought reasoning. These two requirements help the LLM understand the task better and improve the data synthesis.

### A.2.2  STATISTICS OF TEST CASES IN HARDTESTS

We collect a total of 47.1k problems from five direct data sources: Codeforces, AtCoder, Luogu, CodeContests, and TACO. After removing the problems that lack oracle programs and the problems that do not read the input from and write the output to standard I/O, we retain 32.5k problems.

We try to generate test cases for these 32.5k problems. Although we carefully design the test case generation prompt, we are not able to achieve 100% coverage. **In the end, we successfully generate test cases for 26.6k problems, forming the HARDTESTS dataset.**

### A.2.3  FAILURE RATE AND FAILURE REASONS OF HARDTESTGEN

The status distribution of test case generation across the 32.5k problems is shown in Figure 4. Overall, we successfully generated test cases for 81.9% of the problems. The main failure reasons include: 1) all "oracle programs" are in fact erroneous (6.62%), 2) all generated outputs are eliminated by consensus filtering (5.85%), and 3) no valid inputs are generated (3.72%).

We present the distribution of the number of test cases in HARDTESTS in Table **??**, and we can see that the vast majority of problems have a sufficiently large number of test cases.

### A.2.4  HARDTESTS EXAMPLES

**Example 1**

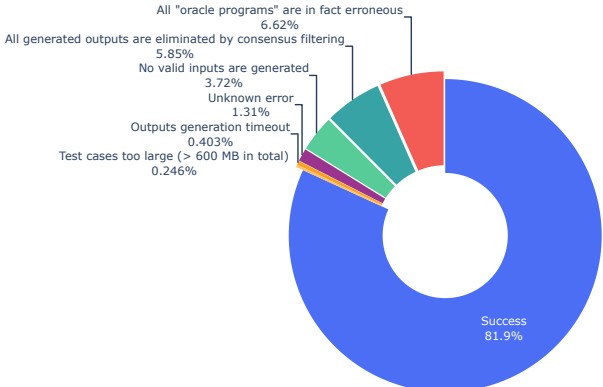

Figure 4: The result status distribution of our test case generation pipeline HARDTESTGEN.

This example demonstrates the input validator, LLMGen and RPGen test cases, as well as a custom judging function. Here's the problem description:

*Codeforces 1096A: There are a total of $T$ ($1 \leq T \leq 1000$) sub-tasks. Each sub-task gives a pair of integers $l, r$ ($1 \leq l \leq r \leq 998244353$), and the goal is to find a pair of integers $x, y$ such that $l \leq x, y \leq r$, $x \neq y$, and $y$ is divisible by $x$. It is guaranteed that every sub-task has a valid solution.*

*Note: It can be mathematically proven that a sub-task has a solution if and only if $2l < r$.*

The input validator is as follows. It checks whether `input_str` conforms to the required format specified in the problem specification, whether all data falls within the required ranges, and whether other constraints are satisfied (e.g., whether each sub-task has a solution).

```python
import sys

def validate_input(input_str: str) -> bool:
    lines = input_str.strip().split('\n')
    if not lines:
        return False

    try:
        T = int(lines[0])
    except:
        return False

    if not (1 <= T <= 1000):
        return False

    if len(lines) != T + 1:
        return False

    for i in range(1, T + 1):
        parts = lines[i].strip().split()
        if len(parts) != 2:
            return False
        try:
            l, r = map(int, parts)
        except:
            return False

        if not (1 <= l <= r <= 998244353):
            return False

```

```
31          if 2 * l > r:
32              return False  # No valid pair possible
33
34      return True
```

Since this problem allows multiple correct solutions, simple string comparison is not sufficient. We need a special, customized output judging function. The output judging function is as follows.

```python
1  def output_judging_function(input_str: str, candidate_output: str, reference_output:
   ↪ str) -> bool:
2      try:
3          input_lines = input_str.strip().split('\n')
4          T = int(input_lines[0])
5          queries = [tuple(map(int, line.strip().split())) for line in input_lines[1:T+1]]
6
7          output_lines = candidate_output.strip().split('\n')
8          if len(output_lines) != T:
9              return False
10
11         for (l, r), line in zip(queries, output_lines):
12             parts = line.strip().split()
13             if len(parts) != 2:
14                 return False
15             x, y = map(int, parts)
16             if not (l <= x <= r and l <= y <= r):
17                 return False
18             if x == y:
19                 return False
20             if y % x != 0:
21                 return False
22
23         return True
24     except:
25         return False
```

The LLMGen inputs are as follows.

```
1  [
2      "3\n1 10\n2 8\n3 10",
3      "2\n5 20\n10 25",
4      "3\n7 30\n1 5\n2 6",
5      "1\n100 300",
6      "2\n999 2000\n1000 3000"
7  ]
```

The RPGen input generator is as follows. To ensure a solution always exists, the LLM sets $r \geq 2l$.

```python
1  import random
2
3  def gen_range_based_input() -> str:
4      T = random.randint(1, 1000)
5      queries = []
6      for _ in range(T):
7          l = random.randint(1, 499122176)
8          r = random.randint(2 * l, 998244353)
9          queries.append(f"{l} {r}")
10     return f"{T}\n" + "\n".join(queries)
```

The LLM believes that there is no need to generate a SPGen input generator or HackGen input generator for this problem.

### Example 2

This example demonstrates the input validator, as well as the LLMGen, SPGen, and HackGen test cases. Here's the problem description:

*Codeforces 1141 A: Given two numbers $n, m$ ($1 \leq n \leq m \leq 5 \times 10^8$), you are to determine whether it is possible to transform $n$ into $m$ by multiplying by 2 and 3, and if so, output the minimum number of operations. Otherwise, output -1.*

The input validator is as follows. It checks whether `input_str` conforms to various format requirements and constraints.

```python
import re

def validate_input(input_str: str) -> bool:
    try:
        # Split the input string into two parts
        parts = input_str.strip().split()
        if len(parts) != 2:
            return False
        # Convert parts to integers
        n, m = map(int, parts)
        # Check the constraints
        if not (1 <= n <= m <= 5 * 10**8):
            return False
        return True
    except:
        return False
```

The LLMGen inputs are as follows.

```python
[
    "120 51840",
    "42 42",
    "48 72",
    "1 1",
    "2 6",
    "3 9",
    "5 10",
    "7 21",
    "8 24",
    "10 30"
]
```

The SPGen generator is as follows. The output of this problem has two categories (i.e., possible and impossible), so the LLM generates two regular input-generating functions, corresponding to these two categories, respectively.

Since this problem requires SPGen, the LLM does not generate any RPGen input generators following the instructions given in the prompt.

```python
import random

def gen_stratified_input_for_category_possible() -> str:
    n = random.randint(1, 10**8)
    m = n
    for _ in range(random.randint(1, 20)):
        if random.choice([True, False]):
            m *= 2
        else:
            m *= 3
        if m > 5 * 10**8:
            break
    return f"{n} {m}"

def gen_stratified_input_for_category_impossible() -> str:
    n = random.randint(1, 10**8)
    m = random.randint(n + 1, 5 * 10**8)
    while m % n == 0:
```

```
19          m += 1
20      return f"{n} {m}"
```

The HackGen generator is as follows. The LLM generates two hacking input generating functions. The first function sets a small $n$ and a large $m$. This is because a brute-force approach that a candidate program might take is to use DFS, recursively trying to multiply $n$ by 2 or 3 until it becomes greater than or equal to $m$. If we randomly choose $n$ and $m$, the ratio between them is usually small, so this approach might still pass. Setting $n$ to be small and $m$ to be big creates a large gap between $n$ and $m$, making the brute-force DFS approach inefficient. The second function sets $m = n$, which serves as an edge case.

```
1   import random
2
3   def gen_hacking_input_for_small_n_big_m() -> str:
4       n = random.randint(1, 5)
5       m = random.randint(4 * 10**8, 5 * 10**8)
6       return f"{n} {m}"
7
8   def gen_hacking_input_for_edge_case() -> str:
9       n = random.randint(1, 5 * 10**8)
10      return f"{n} {n}"
```

For this problem, the LLM believes that a string comparison function would be enough for output judging.

### A.3 DETAILS OF THE COLLECTION OF PROBLEM SPECIFICATIONS AND ORACLE PROGRAMS IN HARDTESTS

We collect 47,136 algorithmic coding problems from five direct data sources: AtCoder, Codeforces, Luogu, CodeContests, and TACO, and these problems are originated from 13 online judge platforms, including Codeforces, AtCoder, and SPOJ.

**Data sources.** *Codeforces* (https://codeforces.com/) is one of the largest English online judge platforms. We collected all publicly available problem specifications up to September 2024 from Codeforces. *AtCoder.* (https://atcoder.jp/) is a large online judge platform offering problems in both Japanese and English. We scraped all problem specifications available up to September 2024, along with three correct user-submitted C++ programs for each problem. We used those directly for problems with official English versions. *Luogu* (https://www.luogu.com.cn/) is a large Chinese online judge platform consisting of a main section (Luogu-Main) and four mirror sections. The main section hosts original problems authored by users and administrators, as well as problems sourced from real-world contests (e.g., USACO). The mirror sections contain problems from other platforms, including AtCoder, SPOJ, Codeforces, and UVa. We collected all available problem specifications and community-authored tutorials, which often include both correct C++ programs and corresponding natural language explanations, from Luogu. *CodeContests* (Li et al., 2022) is a dataset comprising 13,493 problems collected from five platforms. Each entry includes a problem specification and several correct programs in C++, Python 2, Python 3, and Java. Only Codeforces problems in CodeContests were used in our dataset, as only their problem IDs were explicitly provided. *TACO* (Li et al., 2023) is a large-scale English dataset containing 25.4k problems sourced from ten platforms. Each entry includes a problem specification and multiple correct Python programs. We collect all problems from TACO.

The distribution of problem counts across each online judge platform is shown in Figure 5. The URLs of each platform, along with the direct data sources of their problem specifications and oracle programs, are listed in Table 5.

Note that since some problems have multiple oracle program sources, we prioritize programs from more reliable sources when generating test cases. The reliability, supported languages, and notes regarding each direct source of oracle programs are presented in Table 6. The distribution of the number of oracle programs per problem is shown in Figure 6.

**Decontamination Process** We collected problems from five direct data sources (TACO, CodeContests, Luogu, Codeforces, and AtCoder), so did LiveCodeBench.

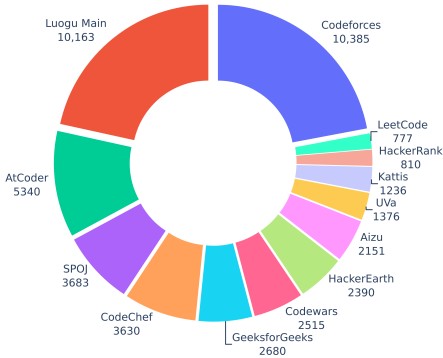

Figure 5: Number of problems from each online judge platform.

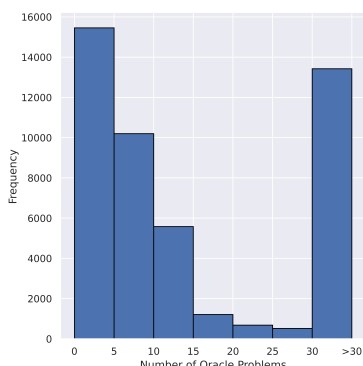

Figure 6: Distribution of the number of oracle programs in HARDTESTS.

Table 5: Problem specification sources and oracle solution sources of each online judge platform.

| platform | URL | Problem Specification Sources | Oracle Program Sources |
|---|---|---|---|
| Codeforces | https://codeforces.com/ | Codeforces | TACO, CodeContests, Luogu |
| AtCoder | https://atcoder.jp/contests/ | AtCoder | AtCoder, TACO, Luogu |
| Luogu | https://www.luogu.com.cn/ | Luogu | Luogu |
| UVa | https://onlinejudge.org/ | Luogu | Luogu |
| SPOJ | https://www.spoj.com/ | Luogu | Luogu |
| Aizu | https://onlinejudge.u-aizu.ac.jp/ | TACO | TACO |
| GeeksforGeeks | https://www.geeksforgeeks.org/ | TACO | TACO |
| Codewars | https://www.codewars.com/ | TACO | TACO |
| Kattis | https://open.kattis.com/ | TACO | TACO |
| CodeChef | https://www.codechef.com/ | TACO | TACO |
| HackerEarth | https://www.hackerearth.com/ | TACO | TACO |
| LeetCode | https://leetcode.com/ | TACO | TACO |
| HackerRank | https://www.hackerrank.com/ | TACO | TACO |

We then compare each problem in our dataset with every problem from LiveCodeBench, using the following criteria to check whether two problems are too similar:

- If two problems come from different source platforms, we convert their problem statements into 3-gram sets and compute their Jaccard similarity (intersection over union). If the score exceeds 0.7, we treat them as the same problem and only keep one.

- If two problems come from the same source platform, e.g. AtCoder, we instead compare their official problem IDs on that platform. We do not apply n-grambased similarity in this case because problem setters on the same platform may slightly modify an existing problem to create a newer and harder version, which can make the n-gram similarity unreliable. In contrast, the platform-assigned problem IDs are much more reliable.

## A.4 EVALUATION RESULT ON ATCODER AND CODEFORCES SEPARATELY

For completeness, we report the evaluation results separately on the AtCoder and Codeforces subsets of the combined evaluation set. Table 12 and Table 13 show the precision and recall of test cases from TACO, CodeContests, and HARDTESTS on LLM-generated programs. Overall, our dataset HARDTESTS consistently improves both precision and recall across both platforms.

Table 6: Oracle program sources with reliability, languages, and notes

| Oracle Program Source | Reliability | Languages | Notes |
|---|---|---|---|
| User-submitted and accepted programs from AtCoder | High | Python, C++ | Some code (either Python or C++) may use AtCoder's custom library. |
| Code solutions from CodeContests | High | Python 2/3, C++, Java | — |
| Community-authored editorials from Luogu | Medium | C++ | Some editorials may lack complete, directly executable code. But if the code has no compilation or runtime errors, it is very likely to be completely correct. |
| Verified programs from TACO, i.e., programs that can pass all TACO's own test cases | Medium | Python | There's some false positives in TACO's test cases. |
| Other programs from TACO | Low | Python | Reliability is not zero due to some false negatives in TACO's test cases. |

Table 7: Precision and recall of the test cases of HARDTESTS on the subset of problems from AtCoder.

| | Difficulty 1 | | Difficulty 2 | | Difficulty 3 | | Difficulty 4 | | Average | |
|---|---|---|---|---|---|---|---|---|---|---|
| | prec. | recall | prec. | recall | prec. | recall | prec. | recall | prec. | recall |
| HARDTESTS (ATCODER) | 97.58 | 98.91 | 88.39 | 100.0 | 69.66 | 88.57 | 53.33 | 100.0 | 77.24 | 96.87 |

## A.5 DETAILED PROTOCOL OF THE DIRECT EVALUATION OF TEST CASES' QUALITY

**Evaluation details for LLM-generated programs on AtCoder.** AtCoder previously made its official test cases publicly available. Although this is no longer the case, we obtained a partial archive from the Github repository `conlacda/atcoder-testcases`. We selected problems that are both in TACO and HARDTESTS, resulting in a total of 653 problems. Since there are almost no AtCoder problems in CodeContests, we generate test cases for these problems by implementing the test case generation procedure described in the CodeContests' paper.

**Evaluation details for LLM-generated programs on Codeforces.** Codeforces does not make its test cases publicly available. Therefore, we manually submit LLM-generated candidate programs to the Codeforces platform to obtain ground-truth verdicts. For each difficulty level from 1 to 4, we randomly select 150 problems that are in TACO, CodeContests, and HARDTESTS at the same time, yielding a total of 600 problems. For problems where the results of all three datasets agree, we randomly select 5% of them for submission. For problems where the datasets produce conflicting results, we submit 50% of the candidate programs. We compute precision and recall based on the combined submission outcomes.

**Evaluation details for human-written programs on Codeforces.** A dataset at Huggingface titled `MatrixStudio/Codeforces-Python-Submissions` collects 690k human-submitted programs on Codeforces along with their official verdicts. We use the verdicts as the ground-truth labels. All other settings are the same as those of evaluation using LLM-generated programs.

**Precision estimates on a larger subset.** We provide precision evaluation on a larger subset of problems. Note that this evaluation does not include comparison with baselines, because many of the problems we evaluate on are not included in the baselines.

We selected all problems in HARDTESTS that originate from AtCoder, have public official test cases, and have a difficulty rating between 1 and 4, resulting in a total of 1.7K problems. We used Qwen2.5-Coder-7B-Instruct to generate five programs for each problem. The results are shown in Table 7, and the performance is consistent with what we reported in the main text.

## A.6 QUALITATIVE ANALYSIS OF GENERATED TESTS

### A.6.1 EXAMPLE 1: FALSE POSITIVE OF TACO AND HARDTESTS LLMGEN

In this example we show how TACO and HARDTESTS LLMGen tests cannot break a wrong program and result in a false positive, while HARDTESTS RPGen tests succeeds in making the program fail. Here's the problem description:

*AtCoder ABC117C: Given an integer $N$ ($2 \leq N \leq 2 \times 10^5$) and an integer array $A$ of length $N$ ($0 \leq A_i \leq 10^9$), compute the value of $\sum_{i=1}^{N-1} \sum_{j=i+1}^{N} A_i A_j$ modulo $10^9 + 7$.*

Since $2 \leq N \leq 2 \times 10^5$, the solution to the problem needs to be relatively efficient. The correct solution employs mathematical techniques to simplify the equation into: $\frac{1}{2} \left( \left( \sum_{i=1}^{N} A_i \right)^2 - \sum_{i=1}^{N} A_i^2 \right)$, which yields an $O(N)$ algorithm.

However, a candidate program generated by Qwen2.5-Coder-7B-Instruct uses a brute-force algorithm with a time complexity of $O(N^2)$. The candidate program is as follows:

```cpp
#include <iostream>
#include <vector>

int main() {
    std::ios::sync_with_stdio(false);
    std::cin.tie(nullptr);

    int N;
    std::cin >> N;

    std::vector<long long> A(N);
    for (int i = 0; i < N; ++i) {
        std::cin >> A[i];
    }

    const int MOD = 1000000007;
    long long sum = 0;

    for (int i = 0; i < N; ++i) {
        for (int j = i + 1; j < N; ++j) {
            sum = (sum + A[i] * A[j]) % MOD;
        }
    }

    std::cout << sum << std::endl;

    return 0;
}
```

Due to its inefficiency, this candidate program failed to pass the official test cases. Nevertheless, because the test cases in TACO and HARDTESTS LLMGen were relatively small (with small $N$), the candidate program successfully passed these cases.

Furthermore, the HARDTESTS RPGen input for this problem is generated using the following Python function:

```python
import random

def gen_range_based_input() -> str:
    n = random.randint(2, 200000)
    a = [random.randint(0, 1000000000) for _ in range(n)]
    return f"{n}\n" + " ".join(map(str, a))
```

Due to the larger scale of HARDTESTS RPGen inputs, the candidate program failed to pass these test cases and we have a true negative.

### A.6.2 EXAMPLE 2: FALSE POSITIVE OF TACO AND HARDTESTS LLMGEN + RPGEN

In this example we show how TACO and HARDTESTS type LLMGen and RPGen tests cannot break a wrong program and result in a false positive, while HARDTESTS HackGen tests succeeds in making the program fail. Here's the problem description:

*AtCoder ABC139C: There are $N$ ($1 \leq N \leq 10^5$) squares arranged from left to right, with the height of the $i$-th square from the left being $H_i$ ($1 \leq H_i \leq 10^9$). Starting from any square, you can move one step to the right as long as the next squares height is not greater than the current one. Find the maximum number of moves possible.*

Given that $1 \leq N \leq 10^5$, the solution needs to be relatively efficient. The correct solution uses an $O(N)$ greedy algorithm. Specifically, it traverses the array from left to right, counting how many consecutive heights satisfy $H_i \geq H_{i+1}$. Each time a condition fails, the counter is reset, and the maximum value is maintained throughout the traversal as the answer.

A candidate program generated by Qwen-Coder-14B-Instruct uses a brute-force approach, iterating over each starting position and simulating the moves. Although functionally correct, its complexity is $O(N^2)$ and too inefficient to work. The code is as follows:

```cpp
#include <iostream>
#include <vector>
using namespace std;

int main() {
    int N;
    cin >> N;
    vector<int> H(N);
    for(int i = 0; i < N; i++) {
        cin >> H[i];
    }

    int max_moves = 0;
    for(int i = 0; i < N; i++) {
        int current_moves = 0;
        for(int j = i + 1; j < N; j++) {
            if(H[j] <= H[j - 1]) {
                current_moves++;
            } else {
                break;
            }
        }
        if(current_moves > max_moves) {
            max_moves = current_moves;
        }
    }

    cout << max_moves << endl;
    return 0;
}
```

Because of its inefficiency, this candidate program failed the official test cases. Nevertheless, due to the relatively small scale of the test cases in TACO and HARDTESTS LLMGen, the candidate program passed these tests.

Additionally, the HARDTESTS RPGen input for this problem is generated using the following Python function:

```python
import random

def gen_range_based_input() -> str:
    N = random.randint(1, 100000)
    heights = [random.randint(1, 1000000000) for _ in range(N)]
    return f'{N}\n' + ' '.join(map(str, heights))
```

We observe that since the $H_i$ sequence is randomly generated, it fluctuates significantly, reducing the complexity of the "simulate moving from a certain square" procedure from $O(N)$ to approximately $O(1)$. Thus, the tests generated do not lead to the worst case complexity of the inefficient program and its overall time complexity effectively becomes $O(N)$, enabling the candidate program to pass HARDTESTS RPGen test cases.

The HARDTESTS HackGen inputs for this problem are generated using the following Python functions:

```python
import random

# Monotonically decreasing sequence
def gen_hacking_input_for_monotonic_decreasing() -> str:
    N = 100000
    heights = list(range(1000000000, 1000000000 - N, -1))
    return f'{N}\n' + ' '.join(map(str, heights))

# Monotonically increasing sequence
def gen_hacking_input_for_monotonic_increasing() -> str:
    N = 100000
    heights = list(range(1, N + 1))
    return f'{N}\n' + ' '.join(map(str, heights))

# Alternating heights
def gen_hacking_input_for_alternating() -> str:
    N = 100000
    heights = [1000000000 if i % 2 == 0 else 1 for i in range(N)]
    return f'{N}\n' + ' '.join(map(str, heights))
```

There are three hacking input generation functions: monotonically decreasing, monotonically increasing, and alternating sequences. The first generated input (monotonically decreasing) successfully increased the actual runtime complexity of the candidate program to $O(N^2)$, causing a timeout and consequently a failure on this test case.

### A.6.3 EXAMPLE 3: FALSE NEGATIVE OF TACO

In this example, we show an example of false negative caused by the lack of output judge function in TACO tests. We also show how HARDTESTS can correctly evaluate the candidate program. Here's the problem description:

*AtCoder ABC117A: Given an integer $T$ and an integer $X$ ($1 \leq T \leq 100$, $1 \leq X \leq 100$). Compute the value of $T/X$ with an error tolerance within $10^{-3}$.*

A candidate program generated by Qwen2.5-Coder-14B-Instruct is:

```cpp
#include <iostream>
#include <iomanip>

int main() {
    int T, X;
    std::cin >> T >> X;

    double timeInWorldA = static_cast<double>(T) / X;

    std::cout << std::fixed << std::setprecision(10) << timeInWorldA << std::endl;

    return 0;
}
```

This is clearly correct and passes all official test cases. It also passes all test cases from HARDTESTS, but it fails on TACOs test cases. This is because using a simple string comparison function is insufficient due to potential differences in precision between the candidate output and the reference output. TACO does not provide a special output judging function for problems, which leads to false negatives. HARDTESTS provides a special output judging function, shown below:

```python
def output_judging_function(input_str: str, candidate_output: str, reference_output:
↪    str) -> bool:
    # Parse the input
    T, X = map(int, input_str.split())

    # Calculate the expected output
    expected_output = T / X

    # Parse the candidate output
    try:
        candidate_value = float(candidate_output.strip())
    except ValueError:
        return False

    # Check the absolute and relative error
    absolute_error = abs(candidate_value - expected_output)
    relative_error = absolute_error / abs(expected_output) if expected_output != 0 else
↪    float('inf')

    # The output is correct if either error is within the tolerance
    return absolute_error <= 1e-3 or relative_error <= 1e-3
```

### A.7   DOWNSTREAM TRAINING AND EVALUATION DETAILS

#### A.7.1   REJECTION SAMPLING TRAINING AND EVALUATION DETAILS

In the rejection sampling experiments, our model is trained with the following training parameters (epochs=20, learning_rate=4e-5, batch_size=128, cosine learning rate schedule with a decay to 10% of the peak learning rate and 32,768 max length). The evaluations are sampled with temperature=0.6, top_p=0.95, top_k=20, min_p=0, max_new_tokens=32768 as recommended by Qwen.

#### A.7.2   RL TRAINING AND EVALUATION DETAILS

We use verl for RL training and firejail for sandboxing code execution. The rollouts are generated with temperature=1, top_p=0.95, top_k=20, min_p=0, response_length=24000, initial learning rate 5e-7. We use a global batch size of 32 and generate 32 samples per rollout. All our experiments are run on 8 NVIDIA H100 GPUs. We do not use KL divergence in our RL loss.

### A.8   TEST CASE GENERATION WITHOUT AN ORACLE MODEL

In the case that an oracle program $y^*$, or an oracle test suite $V^*$ does not exist for a problem $x$, such as when problems are synthetically generated, we propose a method, based on ALGO (Zhang et al., 2023) that synthesizes both the oracle and tests. To start, we prompt an LLM, such as Anthropic Claude 3.5 Sonnet, to generate a brute-force solution $y_{bf}$ to the problem. Specifically, we encourage it to use inefficient methods such as exhaustive search and enumeration of the possible output space. This is founded on the observation that it is relatively easy to generate a solution that exhaustively searches the correct output, but more difficult to optimize it within a time complexity bound.

Then, an LLM is prompted to create a validator program and 10 edge test input generators, which are used to generate one test input each, $\{a_1, \ldots, a_{10}\}$. To prevent the $y_{bf}$ from timing out when computing their respective outputs, we explicitly prompt the LLM to keep input values small. Once these test inputs are verified for correctness using the validator, the brute-force solution is used to generate the corresponding outputs $c_i = y_{bf}(a_i)$ for each input, resulting in a total of 10 input-output pairs as test cases. Finally, the LLM is prompted to create one maximum-length test case $a_{max}$ with inputs at the upper bounds of the problem's constraints, designed to catch solutions that are functionally correct but inefficient. This test case is considered to be passsed as long as the program produces an output before timing out. Crucially, all 11 of the generated test cases $\{a_1, \ldots, a_{10}, a_{max}\}$ are designed to cause seemingly correct programs to fail, and none are generated using random inputs.

We compare this method to the baseline method outlined in AceCoder (Zeng et al., 2025a), which uses a direct prompt to generate 20 full test cases (inputs and corresponding outputs), also using Claude 3.5. Then, after prompting a stronger model such as Qwen2.5- Coder-32B-Instruct to gener-

ate a solution, the test cases that cause the solution to fail are considered hallucinated and are filtered out. Problems with fewer than 5 test cases after filtering are discarded.

To evaluate the accuracy of rewards that our test cases can give to model training, we evaluate the precision and recall over candidate programs generated by LLMs and written by humans on subsets of problems in HARDTESTS.

The quality of the test cases are verified using 165 Atcoder problems, each with 50 sample solutions. It is clear from these experiments (shown in Table 8) that our method can also work much better than the baseline even when oracle programs are not available. The false positive rate of HARDTESTGEN is only half as high as AceCoder, showing that deliberately crafting high-quality, hard test cases is crucial for effective program verifiers.

We will show some examples of the test generation process in the following sections.

Table 8: Performance comparison of oracle-free test generation algorithms based on error rates.

|  | False Positive Rate (FPR) | False Negative Rate (FNR) |
|---|---|---|
| AceCoder | 32.49 | 2.59 |
| HARDTESTGEN | 17.67 | 2.19 |

### A.8.1   EXAMPLE 1: BRUTE-FORCE ORACLE SOLUTION GENERATED BY ALGO

In this example, we show a brute-force oracle solution $y_{bf}$ that ALGO generates. The problem is challenging, and while the LLM is unable to correctly solve it directly, its brute-force solution works. Here's the problem description:

*AtCoder ABC301C: A single-player card game is popular in AtCoder Inc.*
*Each card in the game has a lowercase English letter or the symbol  written on it. There is plenty number of cards for each kind. The game goes as follows.*

*- Arrange the same number of cards in two rows.*
*- Replace each card with  with one of the following cards: a, t, c, o, d, e, r.*
*- If the two rows of cards coincide, you win. Otherwise, you lose.*

*To win this game, you will do the following cheat.*

*- Freely rearrange the cards within a row whenever you want after step 1.*

*You are given two strings S and T ($1 \leq |S|, |T| \leq 2 \times 10^5$), representing the two rows you have after step 1. Determine whether it is possible to win with cheating allowed. If it is possible to win with cheating allowed, print Yes; otherwise, print No.*

The solution generated by Claude 3.5 Sonnet is:

```python
from itertools import product

def solve():
    S = input().strip()
    T = input().strip()

    # Find positions of @ in both strings
    s_at_pos = [i for i, c in enumerate(S) if c == '@']
    t_at_pos = [i for i, c in enumerate(T) if c == '@']

    # Possible characters to replace @
    possible = ['a', 't', 'c', 'o', 'd', 'e', 'r']

    # Try all possible combinations
    for s_comb in product(possible, repeat=len(s_at_pos)):
```

```
16          # Create new S string with replacements
17          s_new = list(S)
18          for pos, char in zip(s_at_pos, s_comb):
19              s_new[pos] = char
20          s_new = ''.join(s_new)
21
22          for t_comb in product(possible, repeat=len(t_at_pos)):
23              # Create new T string with replacements
24              t_new = list(T)
25              for pos, char in zip(t_at_pos, t_comb):
26                  t_new[pos] = char
27              t_new = ''.join(t_new)
28
29              # Sort both strings (since we can rearrange)
30              if sorted(s_new) == sorted(t_new):
31                  print("Yes")
32                  return
33
34      print("No")
35
36  solve()
```

This solution is evidently inefficient. A full analysis yields runtime in the bound of $O(n \log n7^n)$, where $n = |S| = |T|$, but the exponential complexity is clear from the Cartesian product function in the implementation. For inputs $n > 10$, this algorithm quickly becomes intractable. However, for inputs $n \leq 10$ it is able to generate valid test outputs, allowing it to correctly evaluate the validity of submitted solutions. The test outputs it generates achieve a 100% accuracy, compared to actual execution results from the online judge platform.

### A.8.2 EXAMPLE 2: TEST CASES GENERATED BY ALGO

In this example we show a contest coding problem for which ALGO effectively generates a testing suite. Here's the problem description:

*AtCoder cafeteria sells meals consisting of a main dish and a side dish. There are $N$ types of main dishes, called main dish 1, main dish 2, ..., main dish $N$. Main dish $i$ costs $a_i$ yen. There are $M$ types of side dishes, called side dish 1, side dish 2, ..., side dish $M$. Side dish $i$ costs $b_i$ yen.*

*A set meal is composed by choosing one main dish and one side dish. The price of a set meal is the sum of the prices of the chosen main dish and side dish.*

*However, for $L$ distinct pairs $(c_1, d_1)$, ..., $(c_L, d_L)$, the set meal consisting of main dish $c_i$ and side dish $d_i$ is not offered because they do not go well together. That is, $NM - L$ set meals are offered. (The constraints guarantee that at least one set meal is offered.)*

*Find the price of the most expensive set meal offered.*

*The input is given from Standard Input in the following format:*
*$N$ $M$ $L$*
*$a_1$ $a_2$ ... $a_N$*
*$b_1$ $b_2$ ... $b_M$*
*$c_1$ $d_1$*
*$c_2$ $d_2$*
*$\vdots$*
*$c_L$ $d_L$*

*Constraints:*
*- $1 \leq N, M \leq 10^5$*
*- $0 \leq L \leq \min(10^5, NM - 1)$*
*- $1 \leq a_i, b_i \leq 10^9$*

The first 3 edge test input generators created by ALGO are shown below, corresponding to the following test inputs. Note that the values are at the boundaries of the input bounds and follow clearly defined structures.

```
1  ["1 1 0\n1000000000\n1000000000",
2  "10 10 1\n1000 2000 3000 4000 5000 6000 7000 8000 9000 10000\n1000 2000 3000 4000 5000
   ↪  6000 7000 8000 9000 10000\n1 1",
3  "50 50 100\n1000000000 1000000000 1000000000 1000000000 1000000000 1000000000 1000000000
   ↪  1000000000 1000000000 1000000000 1000000000 1000000000 1000000000 1000000000
   ↪  1000000000 1000000000 1000000000 1000000000 1000000000 1000000000 1000000000
   ↪  1000000000 1000000000 1000000000 1000000000 1000000000 1000000000 1000000000
   ↪  1000000000 1000000000 1000000000 1000000000 1000000000 1000000000 1000000000
   ↪  1000000000 1000000000 1000000000 1000000000 1000000000 1000000000 1000000000
   ↪  1000000000 1000000000 1000000000 1000000000 1000000000 1000000000 1000000000
   ↪  1000000000\n1000000000 1000000000 1000000000 1000000000 1000000000 1000000000
   ↪  1000000000 1000000000 1000000000 1000000000 1000000000 1000000000 1000000000
   ↪  1000000000 1000000000 1000000000 1000000000 1000000000 1000000000 1000000000
   ↪  1000000000 1000000000 1000000000 1000000000 1000000000 1000000000 1000000000
   ↪  1000000000 1000000000 1000000000 1000000000 1000000000 1000000000 1000000000
   ↪  1000000000 1000000000 1000000000 1000000000 1000000000 1000000000 1000000000
   ↪  1000000000 1000000000 1000000000 1000000000 1000000000 1000000000 1000000000
   ↪  1000000000 1000000000\n33 36\n5 1\n18 44\n43 12\n5 37\n50 36\n15 14\n10 27\n34 3\n16
   ↪  40\n47 18\n28 14\n9 10\n20 40\n41 8\n4 41\n10 20\n7 21\n37 1\n49 39\n9 3\n14 26\n42
   ↪  18\n41 28\n26 9\n13 21\n39 3\n16 17\n24 30\n17 46\n33 42\n9 42\n34 7\n36 13\n6 36\n33
   ↪  8\n26 2\n43 18\n4 45\n35 23\n39 5\n33 35\n8 45\n34 18\n43 48\n14 12\n17 11\n26 41\n4
   ↪  47\n44 28\n9 9\n50 19\n2 15\n20 5\n9 39\n20 14\n24 39\n5 13\n4 15\n10 39\n48 23\n43
   ↪  36\n21 33\n4 35\n11 13\n46 10\n11 31\n8 44\n2 37\n46 37\n16 39\n39 43\n34 38\n3 50\n7
   ↪  20\n34 13\n34 31\n28 33\n31 4\n22 1\n38 19\n12 9\n6 23\n27 18\n9 4\n11 10\n28 26\n2
   ↪  7\n45 42\n31 43\n39 1\n15 1\n11 19\n12 48\n15 10\n4 7\n45 26\n2 18\n35 15\n33 27"]
```

```python
1  def gen_edge_case_input_1():
2      # Minimum case: N=M=1, L=0
3      return "1 1 0\n1000000000\n1000000000"
4
5  def gen_edge_case_input_2():
6      # Maximum values case
7      N = M = 100  # Using smaller values for practicality
8      L = N*M - 1
9      a = [10**9] * N
10     b = [10**9] * M
11     pairs = [(i, j) for i in range(1, N+1) for j in range(1, M+1)]
12     pairs.pop()  # Leave one combination possible
13
14     result = f"{N} {M} {L}\n"
15     result += " ".join(map(str, a)) + "\n"
16     result += " ".join(map(str, b)) + "\n"
17     for c, d in pairs:
18         result += f"{c} {d}\n"
19     return result.strip()
20
21  def gen_edge_case_input_3():
22      # Only one forbidden pair
23      N = M = 10
24      a = [i*1000 for i in range(1, N+1)]
25      b = [i*1000 for i in range(1, M+1)]
26      result = f"{N} {M} 1\n"
27      result += " ".join(map(str, a)) + "\n"
28      result += " ".join(map(str, b)) + "\n"
29      result += "1 1"
30      return result
```

Also, the generator for the maximum-length test input $a_{max}$ is shown here. It produces a test input where $N = M = 10^5$, which is the upper bound of the problem.

```python
import random

def gen_maximum_edge_case_input():
    N = 100000
    M = 100000
    L = 100000

    # Generate main dish prices close to max value
    main_prices = [random.randint(999999000, 1000000000) for _ in range(N)]

    # Generate side dish prices close to max value
    side_prices = [random.randint(999999000, 1000000000) for _ in range(M)]

    # Generate L unique forbidden pairs
    used_pairs = set()
    forbidden_pairs = []

    # Start with some specific high-value combinations
    for i in range(L):
        while True:
            c = random.randint(1, N)
            d = random.randint(1, M)
            if (c, d) not in used_pairs:
                used_pairs.add((c, d))
                forbidden_pairs.append((c, d))
                break

    # Build the input string
    result = []
    result.append(f"{N} {M} {L}")
    result.append(" ".join(map(str, main_prices)))
    result.append(" ".join(map(str, side_prices)))

    for c, d in forbidden_pairs:
        result.append(f"{c} {d}")

    return "\n".join(result)
```

This test suite effectively achieves 100% accuracy on evaluating submissions, demonstrating that precise test inputs are crucial for oracle-free verifiers.

### A.8.3 EXAMPLE 3: TEST CASES GENERATED BY ACECODER

For the same Atcoder problem as Example A.8.2, AceCoder generates the following 16 test cases with inputs and outputs after filtering. While the LLM implicitly knows to generate edge test cases, shown in the maximal values of $c_i, d_i$, all of the test cases have relatively similar and low values of $M$ and $N$.

```
[{"input": "2 3 3\n2 1\n10 30 20\n1 2\n2 1\n2 3", "output": "31"},
{"input": "2 1 0\n1000000000 1\n1000000000", "output": "2000000000"},
{"input": "1 1 0\n5\n7", "output": "12"},
{"input": "3 3 4\n10 20 30\n5 15 25\n1 1\n2 2\n3 1\n1 3", "output": "55"},
{"input": "5 3 7\n100 200 300 400 500\n100 200 300\n1 1\n1 2\n1 3\n2 1\n2 2\n3 1\n4 1",
↪    "output": "800"},
{"input": "2 2 1\n999999999 999999998\n999999997 999999996\n1 1", "output":
↪    "1999999995"},
{"input": "3 2 2\n5 4 3\n2 1\n1 1\n2 2", "output": "6"},
{"input": "4 3 5\n10 9 8 7\n6 5 4\n1 1\n2 2\n3 3\n4 1\n4 2", "output": "15"},
{"input": "2 4 3\n100 200\n300 400 500 600\n1 1\n1 2\n2 3", "output": "800"},
{"input": "3 3 0\n1 2 3\n4 5 6", "output": "9"},
{"input": "4 2 3\n10 20 30 40\n50 60\n1 1\n2 2\n3 1", "output": "100"},
{"input": "5 2 4\n1 2 3 4 5\n6 7\n1 1\n2 1\n3 1\n4 1", "output": "12"},
{"input": "3 4 6\n100 200 300\n400 500 600 700\n1 1\n1 2\n1 3\n2 1\n2 2\n3 3", "output":
↪    "1000"},
```

```
14  {"input": "2 2 0\n1000000000 999999999\n1000000000 999999999", "output": "2000000000"},
15  {"input": "3 3 3\n100 200 300\n100 200 300\n1 1\n2 2\n3 3", "output": "500"},
16  {"input": "5 5 12\n1 2 3 4 5\n1 2 3 4 5\n1 1\n1 2\n1 3\n2 1\n2 2\n2 3\n3 1\n3 2\n3 3\n4
    ↪  1\n4 2\n5 1", "output": "10"}]
```

These test cases fail to correctly categorize solutions that exceed the problem's time limit. One such example is shown below, which AceCoder falsely categorizes as a positive solution. Compared to Example A.8.2, in which ALGO generated test inputs as large as $N = M = 10^5$, the test cases from AceCoder are no larger than $N = M = 5$, making them unable to break inefficient programs. Without a brute-force reference oracle, and constrained by the requirement of generating input-output pairs simultaneously, the LLM used by AceCoder sticks to simple test cases that it can be confident are correct. Moreover, longer test cases are likelier to contain hallucinations, and get removed by their filtering process. As a result, their test cases are relatively weaker and result in less effective verifiers.

```python
import sys

def main():
    input = sys.stdin.readline
    N, M, L = map(int, input().split())
    a = list(map(int, input().split()))
    b = list(map(int, input().split()))
    incompatible_pairs = set()
    for _ in range(L):
        c, d = map(int, input().split())
        incompatible_pairs.add((c - 1, d - 1))  # Adjusting indices to be zero-based

    max_price = 0
    for i in range(N):
        for j in range(M):
            if (i, j) not in incompatible_pairs:
                max_price = max(max_price, a[i] + b[j])

    print(max_price)

if __name__ == "__main__":
    main()
```

## A.9    ABLATION STUDY ON LLMs FOR TEST CASE GENERATION

In this work, we primarily used GPT-4o to generate test cases for problems in the HARDTESTS dataset. However, our HARDTESTGEN method is compatible with other LLMs, including open-weight models. We experimented with GPT-4o, Claude-4-Sonnet, Kimi-K2 (Team et al., 2025), and Qwen3-Coder (Team, 2025) as test case generators on 500 randomly selected AtCoder problems. The results are shown in Table 9. We observe that using newer LLMs tends to yield better precision and recall.

We also compared HARDTESTGEN with Qwen3-Coder against prior work TACO and SymPrompt with GPT-4 on the same set of problems. The results are presented in Table 10. Although HARDTESTGEN was paired with Qwen3-Coder, an open-weight model that is considered weaker, it still achieved overall higher precision and recall than TACO and SymPrompt. This indicates that HARDTESTGEN is less dependent on strong proprietary LLMs and maintains reasonable performance even under constrained model conditions.

## A.10    REDUNDANCY ANALYSIS OF TEST CASES

In this section, we analyze the redundancy of test cases. We conducted experiments on 500 randomly selected AtCoder problems. By randomly removing a portion of test cases from both HARDTESTS and the official test sets, we measured the precision of the remaining test cases. The results are shown in Figure 7. We found that HARDTESTS exhibits significantly lower redundancy compared to official test cases. For example, when 60% of the test cases were removed, the precision of official test cases decreased by only 0.63%, whereas the precision of HARDTESTS dropped by 5.94%.

Table 9: Precision and recall of HARDTESTGEN when using different LLMs as test case generators. * denotes open-weight models, and HT denotes HARDTESTGEN.

|  | Difficulty 1 | | Difficulty 2 | | Difficulty 3 | | Difficulty 4 | |
| --- | --- | --- | --- | --- | --- | --- | --- | --- |
|  | Precision | Recall | Precision | Recall | Precision | Recall | Precision | Recall |
| HT+GPT-4o | 99.53 | 99.18 | 100.0 | 97.43 | 96.04 | 98.45 | 84.18 | 98.03 |
| HT+Claude-4-Sonnet | 99.48 | 99.86 | 100.0 | 95.70 | 98.28 | 99.35 | 93.21 | 96.86 |
| HT+Kimi-K2* | 99.41 | 99.87 | 98.30 | 97.01 | 98.06 | 99.13 | 87.11 | 98.04 |
| HT+Qwen3-Coder* | 99.47 | 99.14 | 99.62 | 98.88 | 95.20 | 99.13 | 76.83 | 98.82 |

Table 10: Comparison between HARDTESTGEN with Qwen3-Coder and TACO/SymPrompt with GPT-4. HT denotes HARDTESTGEN.

|  | Diffculty 1 | | Diffculty 2 | | Diffculty 3 | | Diffculty 4 | |
| --- | --- | --- | --- | --- | --- | --- | --- | --- |
|  | Precision | Recall | Precision | Recall | Precision | Recall | Precision | Recall |
| SymPrompt+GPT-4 | 98.74 | 98.95 | 92.64 | 90.91 | 81.72 | 90.99 | 28.13 | 93.18 |
| TACO+GPT-4 | 100.0 | 73.06 | 99.75 | 67.29 | 92.74 | 74.08 | 62.07 | 71.05 |
| HT+Qwen3-Coder* | 99.47 | 99.14 | 99.62 | 98.88 | 95.20 | 99.13 | 76.83 | 98.82 |

We also argue that a certain degree of redundancy is acceptable. On average, each problem in HARDTESTS contains 39 test cases. During reinforcement learning training, we adopt the setting that once a candidate program fails on any test case, evaluation is terminated, and the program is marked incorrect. In practice, we found that evaluating a candidate program on a CPU takes only 5.1 seconds on average, which is negligible compared to the time required for LLM rollout and weight updates. Furthermore, in practical settings, multiple CPUs are usually available.

## A.11 HARDTESTGEN COMPARISON WITH SOFTWARE TESTING METHODS

**AFL++** Traditional fuzzing methods like AFL++ aim to generate random and unexpected input to programs to detect vulnerabilities. However, in algorithmic programming, the input is expected to follow the specification, and the goal for the test cases is to identify a correct and efficient program, not to find a safe program. That said, we still use AFL++ to get the fuzzed inputs for comparison with HARDTESTGEN.

As shown in Table 11a, HARDTESTGEN achieves higher precision and recall for the 53 programs we evaluated. We find that the low precision and recall are due to AFL++ generating a large number of invalid inputs. Problem specification is necessary for generating valid inputs. As such, classical

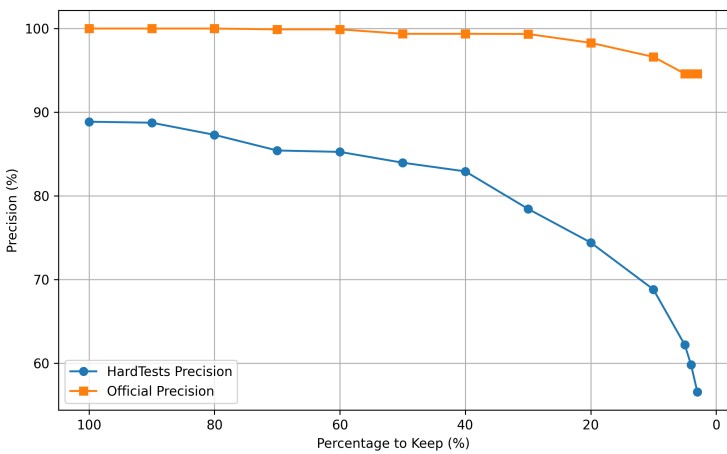

Figure 7: Percentage of retained test cases vs precision.

Table 11: Comparison between HARDTESTGEN and software testing test generation methods.

(a) Comparison between HARDTESTGEN and AFL++.

|  | Precision | Recall |
|---|---|---|
| AFL++ | 70.00 | 30.43 |
| HARDTESTGEN | 100.00 | 86.96 |

(b) Comparison between HARDTESTGEN and TrickCatcher (both using the same model, GPT-4o).

|  | Precision | Recall |
|---|---|---|
| TrickCatcher | 75.76 | 49.50 |
| HARDTESTGEN | 96.43 | 77.88 |

(c) Comparison between HARDTESTGEN and SymPrompt (both using the same model, GPT-4o).

|  | Precision | Recall | F1 | Line Coverage | Branch Coverage |
|---|---|---|---|---|---|
| SymPrompt | 62.28 | 94.67 | 75.13 | 81.59 | 82.84 |
| HARDTESTGEN | 95.77 | 90.67 | 93.15 | 92.83 | 93.36 |

fuzzers would not be suitable in our setting unless paired with custom input mutators. HARDTEST-GEN could be slightly modified to generate input mutators similarly to how we generate input generators. We have not yet explored using HARDTESTGEN to provide custom mutators to classical fuzzers, but this is an interesting direction that we could explore in the future.

**SymPrompt** Although our problem setting is not coverage-guided test generation, because there is not a single program under test. We still adopt and compare HARDTESTGEN with SymPrompt, one of the coverage-guided methods. We use the Oracle program as the focal method for SymPrompt and measure precision, recall, and coverage on 163 GPT-4o-generated candidate programs.

As shown in Table 11c, HARDTESTGEN achieves a slightly lower recall but much better coverage and precision than SymPrompt.

This is because SymPrompt focuses on providing approximate execution paths to LLMs to maximize coverage, while HARDTESTGEN utilizes the LLMs' algorithmic knowledge to generate difficult test cases that make bad algorithms slow.

For example, for the following function that calculates the greatest common divisor with the Euclidean algorithm:

```python
def gcd(a, b):
    if a % b == 0:
        return b
    return gcd(b, a % b)
```

SymPrompt easily generates test cases that reach 100% coverage, but the test cases it generates often get solved in very few recursions. However, HARDTESTGEN recognizes from the problem description and generates a hacking input generator that assigns a and b to be consecutive Fibonacci numbers, which causes the algorithm's worst-case efficiency of $2 \log_2 b + 1$ steps.

While SymPrompt achieves a higher recall, this performance is potentially superficial, arising not from strong test coverage, but from weak and non-discriminative test cases, as suggested by the low F1 value.

To explain this with an extreme case: a test suite that judges any program as correct will have no false negatives, because it won't have any negatives at all. Consequently, its recall will be 100

To concretely illustrate this point and address the reviewers request for an example, we provide a detailed case study on Codeforces Problem 191C Fools and Roads.

The input specifications for this problem are in natural language:

*The first line contains a single integer n ($2 \leq n \leq 10^5$) Each of the next $n - 1$ lines contains two space-separated integers $u_i, v_i (1 \leq u_i, v_i \leq n, u_i \neq v_i)$, that means that there is a road connecting cities $u_i$ and $v_i$. ... The next line contains integer k ($0 \leq k \leq 10^5$) ...*

In short, this problem involves processing a tree with up to $10^5$ nodes and answering up to $10^5$ pairwise path queries. Efficient solutions require tree traversal preprocessing, and naive or brute-force solutions quickly become infeasible at scale.

We observe that SymPrompt-generated test cases for this problem are small across all generations; the maximum observed values are n = 8 and k = 3. These sizes are far below the problems limits and do not stress the performance characteristics of candidate programs. In contrast, HARDTESTGEN deliberately generates test cases with n and k values approaching the specification limit ($10^5$), effectively stress-testing the scalability and efficiency of candidate programs. The comparison between the two test suites is as follows:

Both methods can reject a buggy program with an incorrect answer. SymPrompt correctly accepts the 1 correct solution, but incorrectly accepts all 3 inefficient programs (correct answer but not within time limit). On the other hand, HARDTESTGEN correctly rejects all 3 inefficient programs, though it incorrectly rejects the 1 correct solution due to tight time constraints. This could be easily resolved by slightly lessening the input scale or better matching the CPUs used by the online judge platform. This leads to the observed recall gap: SymPrompt has higher recall because it accepts the correct programs that HARDTESTGEN wrongly rejects (performance limit being too harsh). However, it does so at the cost of substantially lower precision, accepting several clearly incorrect programs.

In fact, none of the SymPrompt-generated test cases induce any timeout error for brute-force programs. HARDTESTS correctly identifies all 36 slow programs among the 163 candidate programs spanning 38 problems, each of which has at least two candidate implementations.

**TrickCatcher** We adapt TrickCatcher to our setting by:

- Selecting one oracle solution as the program under test (PUT) and following TrickCatcher's approach to generate the program variants and the input generator
- Retaining only variants that pass the public test cases (which serve as the "Existing Test Suite" in TrickCatcher Figure 2).
- Evaluating the generated tests on 209 LLM-generated programs, with precision and recall calculated by comparing the TrickCatcher's test cases' judgment with the CodeForces Online Judge Platform's judgment of the program correctness.

As shown in Table 11b, HARDTESTGEN performs better than TrickCatcher in both precision and recall. The gap between the two is as big as 20.67 percentage points for precision and 28.38 percentage points for recall.

We attribute this improvement to our input validator and multiple types of tests, especially HackGen tests.

Table 12: Precision and recall of the test cases of TACO, CodeContests*, HARDTESTS, and ablative baseline on AtCoder. HT–L refers to the results using only the test cases of LLMGen from HARDTESTS. while HT–L+R/S refers to the results using only the test cases of LLMGen in addition to, RPGen, or SPGen from HARDTESTS. The asterisk next to CodeContests indicates that this dataset does not provide test cases for AtCoder problems. We implemented the method described in their paper to generate the test cases.

| | Difficulty 1 | | Difficulty 2 | | Difficulty 3 | | Difficulty 4+ | | Average | |
| | prec. | recall | prec. | recall | prec. | recall | prec. | recall | prec. | recall |
|---|---|---|---|---|---|---|---|---|---|---|
| *Qwen2.5-Coder-7B-Instruct* | | | | | | | | | | |
| TACO | **99.48** | 77.09 | 89.66 | 62.9 | 69.07 | 81.71 | 26 | 86.67 | 71.05 | 77.09 |
| CodeContests* | 95.24 | 93.93 | 64.12 | 67.74 | 52.5 | 76.83 | 13.64 | 100 | 56.38 | 84.63 |
| HT–L | 94.63 | **99.84** | 74.7 | **100** | 42.2 | **89.02** | 9.79 | **93.33** | 55.33 | **95.55** |
| HT–L+R/S | 97.85 | 99.35 | **97.64** | **100** | 74.23 | 87.8 | 56 | **93.33** | 81.43 | 95.12 |
| HARDTESTS | 98.15 | 98.95 | 97.58 | 97.58 | **86.75** | 87.8 | **58.33** | **93.33** | **85.2** | 94.42 |
| *Qwen2.5-Coder-14B-Instruct* | | | | | | | | | | |
| TACO | **99.82** | 78 | 93.24 | 69 | 80.23 | 73.4 | 44.3 | 76.09 | 79.4 | 74.12 |
| CodeContests* | 95.32 | 94.11 | 71.28 | 69.5 | 67.26 | 80.85 | 28.85 | 97.83 | 65.68 | 85.57 |
| HT–L | 96.21 | **99.72** | 77.22 | **100** | 58.9 | **96.81** | 20.18 | 97.83 | 63.13 | **98.59** |
| HT–L+R/S | 97.31 | 99.02 | 94.79 | **100** | 87.5 | **96.81** | 68.18 | 97.83 | 86.95 | 98.42 |
| HARDTESTS | 97.99 | 99.02 | **96.95** | 95.5 | **93.33** | **96.81** | **69.84** | 95.65 | **89.53** | 96.75 |
| *GPT-4o* | | | | | | | | | | |
| TACO | **100** | 73.06 | 99.75 | 67.3 | 92.74 | 74.08 | 63.9 | 72 | 89.1 | 71.61 |
| CodeContests* | 99.51 | 94.1 | 94.04 | 78.42 | 86.4 | 79.88 | 57.14 | 89.33 | 84.27 | 85.43 |
| HT–L | 99.42 | **99.47** | 94.31 | **99.32** | 86.39 | **99.42** | 48.86 | **99.67** | 82.25 | **99.47** |
| HT–L+R/S | 99.53 | 99.18 | 99.82 | 97.6 | **96.04** | 98.45 | 79.62 | 99 | 93.75 | 98.56 |
| HARDTESTS | 99.53 | 99.18 | **100** | 97.43 | **96.04** | 98.45 | **84.48** | 98 | **95.01** | 98.27 |

Table 13: Precision and recall of the test cases of TACO, CodeContests, HARDTESTS, and ablative baseline on Codeforces. HT–L refers to the results using only the test cases of LLMGen from HARDTESTS. while HT–L+R/S refers to the results using only the test cases of LLMGen, RPGen, and SPGen from HARDTESTS.

| | Difficulty 1 | | Difficulty 2 | | Difficulty 3 | | Difficulty 4+ | | Average | |
| | prec. | recall | prec. | recall | prec. | recall | prec. | recall | prec. | recall |
|---|---|---|---|---|---|---|---|---|---|---|
| *Qwen2.5-Coder-7B-Instruct* | | | | | | | | | | |
| TACO | **89.64** | 86.13 | 71.07 | 92.91 | 31.06 | 39.47 | 9.82 | **100** | 50.4 | 79.63 |
| CodeContests | 85.74 | 89.24 | 63.73 | 97.64 | 23.8 | 47.54 | 6.67 | **100** | 44.99 | 83.61 |
| HT–L | 74.03 | **95.45** | 34.9 | **98.82** | 16.12 | **55.61** | 5.24 | **100** | 32.57 | **87.47** |
| HT–L+R/S | 87.61 | **95.45** | 40.7 | **98.82** | 45.2 | **55.61** | 33.75 | **100** | 51.82 | **87.47** |
| HARDTESTS | 87.61 | **95.45** | **93.3** | **98.82** | 48.38 | **55.61** | 50 | **100** | **69.82** | **87.47** |
| *Qwen2.5-Coder-14B-Instruct* | | | | | | | | | | |
| TACO | 80.67 | 87.45 | 83.88 | 81.13 | 53.87 | 73.88 | 25.76 | **100** | 61.05 | 85.62 |
| CodeContests | 79.7 | 95.59 | 79.29 | 86.16 | 46.49 | **91.84** | 18.68 | **100** | 56.04 | 93.4 |
| HT–L | 74.43 | **98.64** | 48.21 | **100** | 40.57 | 82.04 | 13.45 | **100** | 44.17 | **95.17** |
| HT–L+R/S | 80.08 | **98.64** | 57.65 | **100** | 59.37 | 80.41 | **46.58** | 90.8 | 60.92 | 92.46 |
| HARDTESTS | **83.19** | **98.64** | **88.44** | **100** | **67.47** | 80.41 | **46.58** | 90.8 | **71.42** | 92.46 |
| *GPT-4o* | | | | | | | | | | |
| TACO | **99.58** | 80.02 | 95.76 | 81.72 | 89.64 | 74.83 | 62.64 | 78.17 | 86.91 | 78.69 |
| CodeContests | 99.47 | 94.8 | 95.25 | 89.89 | 86.83 | 87.08 | 58.28 | 94.31 | 84.96 | 91.52 |
| HT–L | 98.45 | 97.28 | 94.48 | **98.71** | 80.14 | **89.36** | 45.49 | **100** | 79.64 | **96.34** |
| HT–L+R/S | 98.81 | **98.88** | 95.46 | **98.71** | 89.15 | 88.5 | 73.01 | 96.2 | 89.11 | 95.57 |
| HARDTESTS | 98.8 | 98.2 | **95.66** | **98.71** | **92.73** | 88.5 | **79.82** | 94.31 | **91.75** | 94.93 |
| *Human Submission* | | | | | | | | | | |
| TACO | **96.28** | 88.89 | **91.48** | 81.59 | **75.9** | 78.84 | 62.23 | 73.77 | **81.47** | 80.77 |
| CodeContests | 94.15 | 90.06 | 87.47 | 89.99 | 73.11 | 85.1 | 56.8 | 79.88 | 77.88 | 86.26 |
| HT–L | 83.5 | **95.57** | 69.73 | **95.97** | 54.7 | 93.59 | 42.82 | **91.72** | 62.69 | **94.21** |
| HT–L+R/S | 91.73 | 94.22 | 83.79 | 95.17 | 70.95 | **93.89** | 60.81 | 89.35 | 76.82 | 93.16 |
| HARDTESTS | 93.29 | 94.13 | 85.15 | 95.05 | 73.71 | 93.59 | **64.16** | 89.35 | 79.08 | 93.03 |