# OpenReview forum: "HARDTESTGEN: A High-Quality RL Verifier Generation Pipeline for LLM Algorithmic Coding"
_ICLR.cc/2026/Conference — ICLR 2026 Poster_

### Official Review · Reviewer_UCch · 2025-10-30

**Soundness:** 3
**Presentation:** 3
**Contribution:** 3
**Rating:** 6
**Confidence:** 4

**Summary:**

This paper presents HardTestGen, a pipeline that synthesizes “hard” test cases for algorithmic problems via 4 different prompts, an LLM-prompted validator, and multi-oracle consensus for outputs. Using this LLM powered pipeline, the authors create 26.2k problems and show comparable (at easy level) or higher (at higher difficulties) precision and recall than the existing methods of TACO and CodeContents. They further demonstrate that post-training with RLVR on their tests leads to higher scores than post training with tests generated from an existing method, TACO.

**Strengths:**

* Coding LLMs are a hot and popular topic. The authors do well to point out the motivation, survey related works (especially with the extra offered in the appendix), and show practical impact/results
* Their method is automated, and scalable, resulting in over 26,000 problems.
* They provide good evaluations and ablations into their different prompting techniques.
* Concurrent work is acknowledged, well presented, and well positioned.
* Overall the method seems reproducible as the authors offered all of their prompts in the appendix and oracle program sources.

**Weaknesses:**

* I think the motivation could be rephrased. The authors seem to teeter between problems that have incorrect solutions (the solution is genuinely wrong and fails to pass all test cases), and inefficient solutions (solutions that are correct but can take too long to execute). In the end they group both of these distinct cases and “incorrect solutions” but it’s clear that correctness != inefficient.
* The second paragraph of the introduction goes over a tree example where a naive algorithm works for randomly generated trees but gives the worst time complexity for a tree who is a chain. This isn’t well motivated. For example, quicksort has a worst case time complexity of n^2 yet is or has been the default sorting algorithm used in Java, C++, C#, and Julia. Despite the worst case time complexity it is favored over merge sort which has a better worst case complexity.
* 25% of the problems require checking equality in a method other than string comparison and in these cases an LLM is prompted for a judging function. Then on line 256, the authors offer a set in python as an example. Sets in python have a built in equality operator and presumably an LLM would propose using that. It would have been nice to see in these cases how often do LLMs use simple built in solutions, and how often do they have to write something intricate from scratch.
* The authors say that two of their prompting techniques for test generation, RPGen and SPGen are mutually exclusive yet Table 1 does not include HackGen and does in fact combine these two mutually exclusive settings. Are the results in the table still valid?
* Given that this benchmark cost over $6,000 ($0.23/problem * 26,000 problems) I question whether the method is truly scalable. Arguably, maybe a program but certainly properties/contracts of a program could be expressed as a set of rules and an SMT solver could be used to find edge cases. One could imagine replacing the prompts for RPGen+SPGen+HackGen with a single prompt to generate such rules and then running the SMT solver.
* Qualitative evaluations could use a slight boost. For example, Table 2 shows a perplexing result in which training on good examples leads to worse performance. A deeper dive as to why would have made for interesting auxiliary results.

**Questions:**

* The second bullet in weaknesses leads to an interesting gap in the analysis of their methodology. Of the test cases generated, how often do we end up with tests that are forcing the worst case time complexity? One can argue that generating the fastest code leads to better LLMs but without a deeper understanding, at what point do we cross into hyper optimized code that is no longer easily human readable?
* Why was 90% chosen as the consensus threshold?
* Section 5.1, line 426, was the resulting random sampling of trajectories near 50/50 in terms of correctness?
* Why did you not use KL divergence in your RL loss? Could that be why
* Could you provide a taxonomy of programs that still slip through to guide future research?

---

> ### Author Response · Authors · 2025-11-25
> **Response Part 1/4**
>
> Thank you for your thoughtful review. Before providing a detailed response, we would like to clarify that in competitive programming Online Judge platforms (e.g., Codeforces, AtCoder), problems are explicitly defined in terms of both functional requirements in natural language and performance constraints, typically strict limits on execution time and memory usage. The official test cases are designed to cover worst-case scenarios, edge cases, and average-case random inputs. Only solutions that satisfy both the functional and performance requirements are considered correct and receive full credit, any failure in either dimension would result in the solution being marked incorrect, and no credit will be given.
>
> ### W1: The authors seem to teeter between problems that have incorrect solutions (the solution is genuinely wrong and fails to pass all test cases), and inefficient solutions (solutions that are correct but can take too long to execute). In the end they group both of these distinct cases and “incorrect solutions” but it’s clear that correctness != inefficient.
>
> Thank you for the comment. While we acknowledge the conceptual difference between functional incorrectness and inefficiency, this distinction is less meaningful in our setting because both lead to test failure and are treated equivalently in evaluation such as Online Judge platforms. As noted above, in competitive programming, a solution that either produces incorrect output or exceeds the specified time or memory limits will fail some portion of the test cases and is considered incorrect. For this reason, we group both types of failures under “incorrect solutions” in our experiments, reflecting the definition of correctness used on these platforms. We will update the manuscript to explicitly clarify articulate this distinction.
>
> ### W2: The second paragraph of the introduction goes over a tree example where a naive algorithm works for randomly generated trees but gives the worst time complexity for a tree who is a chain. This isn’t well motivated. For example, quicksort has a worst case time complexity of n^2 yet is or has been the default sorting algorithm used in Java, C++, C#, and Julia. Despite the worst case time complexity it is favored over merge sort which has a better worst case complexity.
>
> Thank you for bringing this up. We used the tree example to illustrate the importance of carefully designed test cases that can trigger timeouts for algorithms that perform well on average but may fail on worst case scenarios.
>
> In competitive programming, as noted above, worst-case behavior is explicitly considered. The problem description explicity includes both the time and space complexity constraints, specifying the expected performance that the code solutions must achieve even under worst-case scenarios. Therefore, we will need carefully designed test cases to enforce solutions to fit within these performance constraints.

---

> ### Author Response · Authors · 2025-11-25
> **Response Part 2/4**
>
> ### W3: 25% of the problems require checking equality in a method other than string comparison and in these cases an LLM is prompted for a judging function. Then on line 256, the authors offer a set in python as an example. Sets in python have a built in equality operator and presumably an LLM would propose using that. It would have been nice to see in these cases how often do LLMs use simple built in solutions, and how often do they have to write something intricate from scratch.
>
>
> We analyzed the line-count distribution of special output judging functions (OJFs), as shown in the table below.
>
> | Number of Lines in OJF | Percentage |
> | ---------------------- | ---------- |
> | 1–6                    | 28.76%     |
> | 7–12                   | 10.22%     |
> | 13–18                  | 10.20%     |
> | 19–24                  | 9.65%      |
> | 25–30                  | 11.25%     |
> | 31–36                  | 10.77%     |
> | 37–42                  | 7.96%      |
> | 43–48                  | 4.72%      |
> | 49–54                  | 2.84%      |
> | 55+                    | 3.63%      |
>
> About 39% of OJFs contain no more than 12 lines, indicating that these problems require relatively simple judging logic (e.g., case-insensitive string comparison, set comparison, or floating-point comparisons with tolerances). In such cases, LLMs can often rely directly on Python’s built-in operations such as set comparison.
>
> In addition, about 30% of OJFs have more than 31 lines, indicating that these problems require much more complex judging logic. For example, a shortest-path problem with complicated constraints may require the output to be any shortest path that satisfies the constraints; thus, the OJF must verify both constraint validity and path optimality. In these cases, the LLM has to write a complex verification logic from scratch.
>
>
>
> ### W4: The authors say that two of their prompting techniques for test generation, RPGen and SPGen are mutually exclusive yet Table 1 does not include HackGen and does in fact combine these two mutually exclusive settings. Are the results in the table still valid?
>
> We thank the reviewer for pointing this out. Yes, the result in Table 1 is still valid. Like we mentioned in the paper, SPGen can be viewed as a hierarchical variant of RPGen where we apply RPGen for each input categories (e.g., Yes or No) to ensure balanced test cases. By "mutually exclusive", we meant that either RPGen or SPGen will be used for a problem.
>
> To avoid any confusion, we will update the table notation from HT–L+R+S to HT–L+R/S, indicating that for each question the test set includes LLMGen plus either RPGen or SPGen, depending on the method used.
>
> ### W5.1 Given that this benchmark cost over 6000 (0.23/problem * 26,000 problems) I question whether the method is truly scalable.
>
> Hardtests is a training dataset specifically designed for doing RL training for LLM coding, although it can be repurpose as a benchmark, the method we propose HardTestGen is designed to scalable generation of verifiers for training. We would argue that the cost of 6000 is modest. Concurrent paper such as rStar-Coder[3] and OpenCodeReasoning[4], prompting properitary models such as GPT-4o and Deepseek-R1 to generate 500K to 700K long reasoning traces as training dataset. The cost of these training dataset is much more significant than Hardtests. Moreover, as we have shown in Appendix A.9 Table 7 (Line 2052-2060), the HardTestGen method is model-agnostic and the performance on open-weight models is similar to that of GPT-4o, making our method scalable.
>
> Moreover, [5] estimates that major AI companies routinely spend tens of millions of dollars on acquiring, licensing, and curating high-quality training data (often exceeding their compute cost). In comparison, a per-problem cost of $0.23 is negligible at the scale of modern LLM data budgets.

---

> ### Author Response · Authors · 2025-11-25
> **Response Part 3/4**
>
> ### W5.2: maybe a program but certainly properties/contracts of a program could be expressed as a set of rules and an SMT solver could be used to find edge cases. One could imagine replacing the prompts for RPGen+SPGen+HackGen with a single prompt to generate such rules and then running the SMT solver.
>
> We agree that SMT solvers can be powerful, however, it is infeasible to use formal methods to validate algorithmic problems in our setting. Formal specifications do not exist for most algorithmic problems, especially at the scale of 26.6k. To create such specifications requires huge efforts and domain expertise. Even if they were created, the validity of the specifications themselves would remain questionable, resulting in a scenario of "who verifies the verifier".
>
> We would appreciate it if the reviewer could point us to any previous work that formally validate inputs for algorithmic problems at scale, because existing evidence suggests otherwise.
>
> LLMs are much worse on auto-formalization and formal verification than regular code generation. As reported in AlphaVerus [1], GPT-4o can only generate formal specification and formally verify 27.1% of the problems in HumanEval with a large budget of 256 samples. On the contrary, it can correctly solve 90% of the problems in Python according to [2] and generate correct input validators for 94% with only one sample, according to our experiment.
>
> ### W6: Qualitative evaluations could use a slight boost. For example, Table 2 shows a perplexing result in which training on good examples leads to worse performance. A deeper dive as to why would have made for interesting auxiliary results.
>
> Regarding Table 2, we would like to clarify that although the pass@1 metric drops by ~2.5%, the pass@10 metric increases by ~4%. This indicates that the fine-tuned model generates more diverse and higher-quality candidate solutions overall. Most models are trained and evaluated primaily on Python code, we are conducting the training and evaluation on C++ code which may shift the distribution and cause the performance flutuation in pass@1. In fact, manually looking at some of the failure cases, a lot of the code solutions are indeed failing because of C++ syntax errors (39% of the errors).
>
> ```
> #include <iostream>
> #include <vector>
> #include <algorithm>
> #include <cmath>
> #include <limits>
>
> using namespace std;
>
> int main() {
>     some logic ....
>     cout << fixed << setprecision(15) << min_time << endl;
>     return 0;
> }
>
> Compiler Output:
> /tmp/tmp56xwi9dr.cpp: In function ‘int main()’:
> /tmp/tmp56xwi9dr.cpp:71:22: error: ‘setprecision’ was not declared in this scope
>    71 |     cout << fixed << setprecision(15) << min_time << endl;
>       |                      ^~~~~~~~~~~~
> ```
>
> ### Q1: The second bullet in weaknesses leads to an interesting gap in the analysis of their methodology. Of the test cases generated, how often do we end up with tests that are forcing the worst case time complexity? One can argue that generating the fastest code leads to better LLMs but without a deeper understanding, at what point do we cross into hyper optimized code that is no longer easily human readable?
>
> We would first like to clarify the intuition behind HackGen test cases. Many flawed programs fail either because they do not correctly handle certain corner cases (e.g., an input sequence of length 0) or because they implement unreasonably inefficient algorithms. However, if we rely solely on RPGen/SPGen to randomly generate inputs, these generators may fail to cover specific corner cases, may not simultaneously place all values at their maximal scale, or may not produce the special structural patterns required to expose worst-case behavior. Therefore, certain inputs must be constructed deliberately. For this reason, we cannot directly quantify a single category of test cases exclusively designed to “force worst-case time complexity,” as HackGen is fundamentally aimed at exposing various failure modes of flawed programs.
>
> We agree that readability is an important factor for interpretability and maintainability. In our current settings, we focus on evaluating functional correctness and performance with respect to the problem specification, where readability is not explicitly considered. In fact, many human submissions that optimize for execution time often include less readable “tricks” such as manual loop unrolling and bitwise manipulations, which are harder to interpret than straightforward, readable code.
>
> We argue that these less readable but performance-oriented techniques are still valuable for LLMs to learn, especially in contexts where efficiency is prioritized over readability. That said, incorporating a reward for readability in conjunction with execution rewards is an interesting direction for future work, where one could jointly optimize code readability and performance while ensuring the functional correctness.

---

> ### Author Response · Authors · 2025-11-25
> **Response Part 4/4**
>
> ### Q2: Why was 90% chosen as the consensus threshold?
>
> We note that even though we already employ an input validator, the constraints in some problems may still be too complex to fully enforce, resulting in certain invalid inputs not being filtered out. The outputs corresponding to these invalid inputs are undefined. Therefore, even if two oracle programs are both fully correct, they may produce different outputs on those invalid inputs. For this reason, we set the consensus filtering threshold to 90% instead of 100%.
>
> In addition, if this threshold were set too low, we might end up using oracle programs that are actually incorrect. While the vast majority of the oracle programs we collected are reliable, not all of them are perfectly correct (see Table 5 for a qualitative analysis). This could ultimately degrade the overall quality of the test cases. Therefore, we adopt 90% as the threshold.
>
>
> ### Q3: Section 5.1, line 426, was the resulting random sampling of trajectories near 50/50 in terms of correctness?
>
> The resulting random sampling is purely random in terms of correctness because we want to mimic the case where we do not have a reliable verifier. That said, we also looked at the distribution of correct and incorrect samples of the trajectories, it contains approximately 1/3 incorrect and 2/3 correct trajectories (as determined by HardTests) for all difficulty levels.
>
> ### Q4: Why did you not use KL divergence in your RL loss? Could that be why
>
> Multiple LLM RL papers such as DAPO [6] suggest that removing the KL divergence term can lead to better performance, which was also what we observed and why we did not use KL divergence.
>
> ### Q5: Could you provide a taxonomy of programs that still slip through to guide future research?
>
> In the programs generated by the Qwen2.5-Coder-Instruct-7B model for AtCoder problems, we manually analyzed 10 false-positive cases on HARDTESTS. **We found that the existing taxonomy (e.g., LLMGen, RPGen, SPGen, and HackGen) is already sufficient to cover them.** The primary issue lies in the LLM hallucinating when constraints become complex, which leads to incorrect implementations.
>
> For example, problem ABC177E requires determining whether $N$ given integers are pairwise coprime, setwise coprime (i.e., the GCD of all numbers is 1), or neither. The LLM correctly recognized that SPGen should include one input generation function for each of the three cases. However, it failed to implement them correctly. For instance, for the setwise-coprime case, its strategy was to repeatedly regenerate values until the GCD becomes 1, which was extremely inefficient and leads to timeout.
>
> LLMs may even make simple mistakes. In ABC133C, the task is: given $0 \le L \le R \le 2\times 10^9$, find integers $i, j$ such that $L \le i \le j \le R$ and $(i \times j) \bmod 2019)$ is minimized. The LLM correctly realized that HackGen should maximize the difference between $L$ and $R$, but it only set $R = L + 10001$ for unknown reasons.
>
> Therefore, we believe a promising future direction is to leverage long-thinking models or generate multiple candidates and have the LLM select the best one, in order to further enhance the performance of the HARDTESTGEN pipeline.
>
>
> **Reference**
>
> [1] Aggarwal, et al. "AlphaVerus: Bootstrapping Formally Verified Code Generation through Self-Improving Translation and Treefinement." ICML 2025.
>
> [2] Hurst, et al. "Gpt-4o system card."
>
> [3] Liu, Yifei, et al. "rStar-Coder: Scaling Competitive Code Reasoning with a Large-Scale Verified Dataset."
>
> [4] Ahmad, Wasi Uddin, et al. "Opencodereasoning: Advancing data distillation for competitive coding."
>
> [5] Kandpal, Nikhil, et al. "Position: The Most Expensive Part of an LLM should be its Training Data."
>
> [6] Yu, Q., Zhang, Z., Zhu, R., Yuan, Y., Zuo, X., Yue, Y., Dai, W., Fan, T., Liu, G., Liu, J., Liu, L., Liu, X., Lin, H., Lin, Z., Ma, B., Sheng, G., Tong, Y., Zhang, C., Zhang, M., Zhang, R., Zhang, W., Zhu, H., Zhu, J., Chen, J., Chen, J., Wang, C., Yu, H., Song, Y., Wei, X., Zhou, H., Liu, J., Ma, W.-Y., Zhang, Y.-Q., Yan, L., Wu, Y., & Wang, M. (2025). DAPO: An open-source LLM reinforcement learning system at scale. In Proceedings of the Thirty-Ninth Annual Conference on Neural Information Processing Systems.

---

### Official Review · Reviewer_RVF6 · 2025-10-31

**Soundness:** 3
**Presentation:** 3
**Contribution:** 3
**Rating:** 6
**Confidence:** 2

**Summary:**

This paper addresses the low-quality test case problem in algorithmic coding benchmarks used for LLM training and evaluation.  The authors propose HARDTESTGEN, a multi-stage LLM-based test synthesis pipeline that produces diverse and valid test cases through four modules: LLMGen, RPGen, SPGen, and HackGen.  These modules generate inputs covering functionality, scalability, categorical balance, and adversarial edge cases. Each generated input is validated by an LLM-written input checker and cross-verified with multiple oracle programs for ground-truth outputs, with consensus filtering to ensure reliability. Using HARDTESTGEN, the authors build HARDTESTS, a dataset of 26.6k problems from 13 competitive programming platforms.  Experiments on AtCoder and Codeforces (1,253 problems) and 800 human submissions show substantial improvements in precision and recall over prior datasets (TACO, CodeContests).  Additional studies demonstrate that test quality positively affects downstream post-training tasks such as rejection sampling and reinforcement learning, with notable performance gains on LiveCodeBench-105.

**Strengths:**

- The modular design of HARDTESTGEN (LLMGen, RPGen, SPGen, HackGen) captures diverse failure modes, a broad coverage rarely seen in prior test-synthesis systems. The inclusion of an input validator and consensus output filtering via multiple oracle programs substantially improves validity and robustness.

- HARDTESTS fills an important gap for competitive-programming-level test data. The evaluation on 26.6k problems and the analysis of precision/recall against established datasets is rigorous. The 11-point improvement in both precision and recall and detailed ablations support the paper’s claims.

- The authors go beyond static metrics and show that better verifiers translate into improved LLM training outcomes, especially in reinforcement-learning settings—an important and underexplored direction.

- Method descriptions, example prompts, ablations, and cost statistics (≈ $0.23 per problem) are detailed.

**Weaknesses:**

- HARDTESTGEN assumes access to multiple human-written correct programs to compute reference outputs and validate tests. This assumption limits scalability to unseen or synthetic problems without oracles. While the authors briefly mention an “oracle-free” variant in Appendix A.8, it is minimally evaluated (165 problems) and shows time-out issues. This weakens claims of general applicability.

- The “DOWNSTREAM EFFECTS” section evaluates only Qwen3-4B, and LiveCodeBench-105 contains just 105 questions. This small testbed cannot robustly demonstrate generalization across architectures or domains. Additional experiments on open-weight models (e.g., DeepSeek-Coder, StarCoder2) or larger LiveCodeBench subsets would strengthen the results.

- Although decontamination with LiveCodeBench is mentioned, it is unclear whether other datasets (e.g., CodeContests+, TACO) share problems or reasoning traces with HARDTESTS, which could inflate evaluation gains.

- While quantitative metrics are thorough, the paper could benefit from qualitative examples showing how HARDTESTGEN discovers subtle inefficiencies or corner-case failures that baselines miss.

**Questions:**

1. Can HARDTESTGEN operate when no oracle exists (e.g., synthetic benchmarks or new competition tasks)? What proportion of problems in the dataset had multiple oracles versus single or none?

2. How do the results change if GPT-4o is replaced with a smaller or open-source model (e.g., DeepSeek-Coder-33B)?  Appendix A.9 is mentioned but not summarized.

3. What criteria were used to select the 105-problem subset in LiveCodeBench-105, and how does it differ from the full benchmark?

4. Given the cost of $0.23 per problem, is HARDTESTGEN cost-effective compared to simpler mutation-based generation (e.g., CodeContests+) when scaled to millions of tasks?

---

> ### Author Response · Authors · 2025-11-25
> **Response Part 1/2**
>
> We sincerely thank you for recognizing the novelty of HardTestGen, its coverage advantages, and the thoroughness of our experiments and presentation. Below we respond to the weaknesses and questions you raised.
>
> We would first like to clarify the problem collection and filtering pipeline.
> * As described in Appendix A.2.2 and A.3, we began by collecting all problems from TACO and CodeContests, and we additionally scraped a 27.4K problems from Luogu, Codeforces, and AtCoder. After deduplication, we obtained 47K unique problems.
> * We then removed 6.8K problems that do not use the standard I/O format, and another 7.8K problems that do not have any oracle programs. This left a total of 32.5K problems.
> * Due to issues such as poor-quality oracle programs, we are not able to achieve 100% coverage. In the end, we successfully generated test cases for 26.6K problems, forming the HARDTESTS dataset.
>
> ### W1 & Q1: HARDTESTGEN assumes access to multiple human-written correct programs to compute reference outputs and validate tests. This assumption limits scalability to unseen or synthetic problems without oracles. While the authors briefly mention an “oracle-free” variant in Appendix A.8, it is minimally evaluated (165 problems) and shows time-out issues. This weakens claims of general applicability... Can HARDTESTGEN operate when no oracle exists (e.g., synthetic benchmarks or new competition tasks)? What proportion of problems in the dataset had multiple oracles versus single or none?
>
>
> As noted above, all 26.6K problems included in HARDTESTS have at least one oracle program. Across the full set of 47.1K collected problems, 83.4% have oracle programs. The detailed statistics are:
>
> |              | total | w/o oracles | w/ 1 oracle | w/ 2+ oracles |
> | ------------ | ----- | ----------- | ----------- | ------------- |
> | All problems collected | 47.1K | 7.8K        | 2.7K        | 36.6K         |
> | HARDTESTS    | 26.6K |  0  | 1.6K       |   25.0K      |
>
> ### W2 and Q3: The "DOWNSTREAM EFFECTS" section evaluates only Qwen3-4B, and LiveCodeBench-105 contains just 105 questions. ... What criteria were used to select the 105-problem subset in LiveCodeBench-105, and how does it differ from the full benchmark?
>
> LiveCodeBench-105 is the LiveCodeBench v5 subset with problems that takes stdin as test inputs from Sep 2024 to Jan 2025. It is choosen as the evaluation using stdin is more robust, and that using a newer subset would be less prone to contamination. LiveCodeBench is a widely used benchmark for evaluating LLM's coding capability in competitive programming, with models commonly reporting the metrics on the latest subset for the purpose of decontamination. We are focusing on algorihtmic coding and would argue that it is not designed to claim, although possible, that we can generlize across domains (e.g. software engineering). Due to resource limitations (training Qwen3-4B requires 8 H100s for 3000 GPU hours, costly ~$6000), we only ran experiments on Qwen models that 's around 4B parameters. We hope the performance gain from precision-recall, and training experiments of Qwen3-4B would motivate the use of HardTests and HardTestGen on other models in the research community.
>
> We ran evaluation on a newer LiveCodeBench subset v6 and present the results below for the combined subset (187 questions in total). In general, the performance gain from HardTestGen observed is in similar extent as to the performance in LiveCodeBench-105.
>
> | Model| pass@1 |
> |------|--------|
> |Qwen2.5-7B-Ins|19.46|
> |Olympic-Coder|29.14|
> |Qwen2.5-7B-Ins + HARDTESTS|34.39|

---

> ### Author Response · Authors · 2025-11-25
> **Response Part 2/2**
>
> ### W3: Although decontamination with LiveCodeBench is mentioned, it is unclear whether other datasets (e.g., CodeContests+, TACO) share problems or reasoning traces with HARDTESTS, which could inflate evaluation gains.
>
>
> First, there is a misunderstanding we would like to clarify. CodeContests is a dataset released by DeepMind in Feb 2022 and is used in our paper as a baseline. However, CodeContests+ is a concurrent work released in June 2025.
>
> The overlap among HARDTESTS, CodeContests, and TACO is as follows:
>
> |              | HARDTESTS | CodeContests | TACO  |
> | ------------ | --------- | ------------ | ----- |
> | HARDTESTS    | 26.6K     | 6.8K       | 12.7K  |
> | CodeContests | —         | 13.5K        | 7.5K  |
> | TACO         | —         | —            | 25.4K |
>
> For the results in Table 1, as stated in Section 4.2, the test set consists of 600 Codeforces problems and 653 AtCoder problems. To ensure fair comparison:
>
> * All Codeforces problems are present in HARDTESTS, CodeContests, and TACO.
> * All AtCoder problems are present in HARDTESTS and TACO.
> * Since CodeContests does not contain any AtCoder problems, we manually implemented the CodeContests methodology to generate test cases for the AtCoder subset.
>
> Regarding reasoning traces, the short answer is **no**.
>
> HardTests, CodeContests, and TACO are all training datasets that were decontaminated against LiveCodeBench, which is the test data.
> Whether HardTests have overlap with CodeContests or TACO is irrelevant because having overlap won't affect the performance on LiveCodeBench.
> We believe that it is unlikely that our experiment setting regarding reasoning traces would cause evaluation inflation.
>
> ### W4: While quantitative metrics are thorough, the paper could benefit from qualitative examples showing how HARDTESTGEN discovers subtle inefficiencies or corner-case failures that baselines miss.
>
>
> Appendix A.6 provides detailed qualitative analysis of generated tests. Specifically, we include:
>
> * one example showing a false positive in TACO and HardTestGen-LLMGen,
> * one example showing a false positive in TACO and HardTestGen-LLMGen+RPGen,
> * one example showing a false negative in TACO.
>
> In all three examples, the test cases generated by the full HardTestGen pipeline produces the correct prediction.
>
> ### Q4: Given the cost of $0.23 per problem, is HARDTESTGEN cost-effective compared to simpler mutation-based generation (e.g., CodeContests+) when scaled to millions of tasks?
>
> As noted in our response to Q2, HardTestGen does not rely on expensive GPT-4o and can use significantly cheaper open-source models with similar performance. Therefore, scaling to millions of tasks does not introduce prohibitive cost.
>
> Furthermore, the mutation-based approach used in CodeContests suffers from three key limitations:
>
> * It relies on public test cases, which are not always available.
> * Public test cases are typically few (1–3) and very small, so mutated test cases remain small and fail to catch efficiency issues. In RL training, this leads models to learn brute-force but inefficient solutions.
> * It uses string matching for output equivalence, which fails on tasks requiring special output judging logic, leading to low recall.
>
> In contrast, HardTestGen avoids all these issues and achieves high precision and recall across all difficulty levels, making it far more suitable for downstream post-training.
>
> Moreover, [1] estimates that major AI companies routinely spend tens of millions of dollars on acquiring, licensing, and curating high-quality training data (often exceeding their compute cost). In comparison, a per-problem cost of $0.23 is negligible at the scale of modern LLM data budgets.
>
> **Reference**
>
> [1] Kandpal, Nikhil, et al. "Position: The Most Expensive Part of an LLM should be its Training Data."

---

### Official Review · Reviewer_mZFD · 2025-11-01

**Soundness:** 3
**Presentation:** 3
**Contribution:** 3
**Rating:** 8
**Confidence:** 3

**Summary:**

This paper presents a pipeline for generating synthetic programming questions based on seed questions collected from programming contests. It provides a set of 26,000 generated questions for use in LLM training and shows that training on these questions outperforms training on two other synthetic code datasets in fine-tuning and RL contexts.

**Strengths:**

- The paper compares post-training performance to other synthetic code datasets
- The pipeline seems sensible for using consensus, judging, and automated methods to improve question quality

**Weaknesses:**

- a comparison against a comparable real data-set is not performed
- discussion of how many generated instances were rejected at various stages was not discussed, which would be interesting to know for judging the effectiveness of each component. In addition, without this it is entirely unclear what sort of cost and computational resources would be needed to replicate this study.
- Reviewing a sample to estimate quality does not appear to have been done.

**Questions:**

- I'm curious what impact this has on less in-distribution coding, for example SWE-bench

- Agreeing on 90 of inputs doesn't seem like a lot, I'm somewhat surprised this doesn't lead to quite a few errors in tests.
"If two oracle programs generate outputs that are equivalent on more than 90% of the inputs (i.e., semantically the same rather than strictly identical), we regard this agreement as valid. "

---

> ### Author Response · Authors · 2025-11-25
> **Response**
>
> Thank you very much for recognizing the quality of our work. This truly means a lot to us. Below, we address the weaknesses and questions you raised.
>
> ### W1: Comparison against a comparable real dataset is not performed.
>
> When conducting this work, the well-known large-scale datasets in competitive programming were APPS (10K problems), TACO (25.4K problems), and CodeContests (13.5K problems). Therefore, we selected the two larger datasets, TACO and CodeContests, as baselines. In terms of scale, both are comparable to our HARDTESTS dataset (26.6K problems). Moreover, their problems originate from real platforms (e.g., Codeforces), and they include their own test case generation methods. For these reasons, we consider TACO and CodeContests to be comparable real datasets.
>
> ### W2: Discussion of how many generated instances were rejected at various stages was not provided, which would be interesting to judge the effectiveness of each component. In addition, without this it is unclear what computational resources would be needed to replicate this study.
>
> As described in Appendix A.2.3, we attempted to generate test cases for 32.5K problems. Among them, 3.72% had no valid inputs generated at all; 7.02% could not produce any outputs because the "oracle programs" were erroneous (e.g., runtime errors or timeouts); 5.85% produced outputs that were all filtered out during consensus filtering; and 1.56% failed for other reasons. As a result, only 81.9% of problems (26.6K) successfully produced test cases, forming the final HARDTESTS dataset.
>
> In addition, as noted in Section 3.4, we used GPT-4o to generate input validators, generators, and output judge functions, resulting in an average API cost of 0.23 USD per problem. Note that such cost is negligible compared to the data budgets typically reported for modern LLM training [1]. However, as shown in Appendix A.9, open-source models such as Kimi-K2 and Qwen3-Coder can achieve similar performance, further reducing cost. We will also release all of these artifacts alongside the dataset.
>
> Moreover, Appendix A.10 shows that generating test cases for each problem takes 5.1 seconds on average on a single CPU, indicating strong scalability. Overall, the computational resources required to replicate this study are modest.
>
> ### W3: Reviewing a sample to estimate quality does not appear to have been done.
>
> For concrete examples, Appendix A.6 provides detailed qualitative analyses of generated tests, including:
>
> - an example showing a false positive in TACO and HardTestGen-LLMGen,
> - an example showing a false positive in TACO and HardTestGen-LLMGen+RPGen,
> - an example showing a false negative in TACO.
>
> In all three examples, the full HardTestGen pipeline produces the correct outcome.
>
> For an overall quality estimate, we manually checked the input generator, input validator, and output judging functions for 100 problems for correctness. We found that 95%, 94%, and 92% of them are correct respectively.
>
> ### Q1: I am curious what impact this has on less in-distribution coding, for example SWE-bench.
>
> Thank you for the question! Training LLMs on competitive programming problems improves their complex reasoning and programming capabilities. Compared to baselines like TACO and CodeContests, HARDTESTS, with more reliable test cases and a larger scale, can help researchers achieve stronger post-training results (e.g., reject sampling and RL). We expect that training on our dataset would also improve performance on coding benchmarks such as HumanEval or SWE-bench. However, since the core contribution of this work is improving test case reliability for competitive programming to support training, rather than studying cross-domain transfer to SWE tasks, we did not run experiments specifically targeting this.
>
> ### Q2: Agreeing on 90% of inputs does not seem like a lot. I am somewhat surprised this does not introduce many test errors. "If two oracle programs generate outputs that are equivalent on more than 90% of the inputs (i.e., semantically the same rather than strictly identical), we regard this agreement as valid."
>
> We set the threshold to 90%, rather than higher, because we note that even though we already employ an input validator, the constraints in some problems may still be too complex to enforce perfectly, leaving some invalid inputs unfiltered. The outputs on such invalid inputs are undefined. As a result, even two fully correct oracle programs may produce different outputs on those invalid cases. For this reason, we adopt 90% as the consensus filtering threshold.
>
>
> **Reference**
>
> [1] Kandpal, Nikhil, et al. "Position: The Most Expensive Part of an LLM should be its Training Data."

---

### Official Review · Reviewer_Ksf8 · 2025-11-06

**Soundness:** 2
**Presentation:** 2
**Contribution:** 3
**Rating:** 4
**Confidence:** 4

**Summary:**

Hardtestgen is proposed as an LLM pipeline to generate verifiers for coding problems, with the key goal to achieve higher quality (precision and recall) verifiers than datasets from prior work. The core improvement is brought by augmenting the test generation with diverse test inputs and edge cases. This is accomplished by using LLM generated programs to generate test cases instead of generating test cases directly with an LLM. The resulting dataset is shown to be higher in quality than existing tests besides being larger in scale. When used to post-train code LLMs these proposed verifiers show higher effectiveness due to their quality and lead to stronger model performance.

**Strengths:**

- The space of datasets that provide execution-based feedback for RL algorithms is limited; this work is well-motivated and can advance research on training of stronger code LLMs.
- The analysis of current datasets’ limitations is fresh and under-addressed in prior work. The critique of prior test-suite generation methods is valid; the proposed improvement - explicitly generating diverse inputs and edge cases - is solid and novel.
- Even though only briefly discussed, the investigation into how verifier/test quality affects LLM post-training offers valuable insights.

**Weaknesses:**

- Precision estimates are based on a small subset. The paper tackles dataset quality but reports precision using <5% of the full set (~1.2k of 26k), which - while better than prior work - does not convincingly characterize the overall dataset quality.
- Verifier quality beyond correctness is not analyzed. Apart from precision/recall, the cost profile of the verifier is missing. In particular, the execution time of HARDTESTGEN’s test suites relative to prior work is not discussed; a costlier suite can be problematic. Ideally, we would prefer a minimal test count that still accurately verifies correctness, but this dimension of test-suite quality is not addressed.
- Limitations are under-discussed. The gap in precision raises questions: when a test suite is found to be incorrect (FP or FN), is that suite still retained in the final 26.6k-problem dataset? Even if improvements over prior work hold, precision in the 80% or 50–60% ranges (Figure 1) could inject noisy signals into post-training.
- Cascaded LLM components lack evaluation. The overall pipeline is a cascade of LLM calls (Sec 3.2/3.3), each prone to error, yet individual components are not sufficiently analyzed. For instance, the LLM-generated output-equivalence function (discussed around L259) needs a reported correctness rate; at minimum, a manually inspected subset could establish its reliability.

**Questions:**

- How does the quality of the generated datasets vary with stronger frontier models (which have moved beyond GPT-4o in coding/reasoning)?
- Filtering in 3.4. What percentage/number of test cases are filtered out? Have you tried automatic re-attempts (regeneration) to raise precision/recall when a generated suite is flagged as incorrect?
- Decontamination in 3.5. How exactly is decontamination performed (e.g., exact-match criteria, near-duplicate thresholds)?
- In Table 1: Why are precision/recall gains on human submissions smaller than on model-generated programs?
- Can you explain the rejection-sampling study design? A more informative experiment would be similar to Figure 3 with an A/B setup for rejection sampling: (A) use HARDTESTS verification feedback vs. (B) use TACO verification feedback, to isolate the impact of verifier quality in that setting.

---

> ### Author Response · Authors · 2025-11-25
> **Response Part 1/3**
>
> Thank you for your thoughtful review and for recognizing the novelty and robustness of our work. This truly means a lot to us. Below, we address the weaknesses and questions you raised.
>
> ### W1: Precision estimates are based on a small subset.
>
> **Our precision estimates are necessarily limited to problems where ground truth data is available and overlap exists with baseline datasets.**
>
> Estimating precision requires comparing our *generated test verdicts* against *ground truth verdicts*, which necessitates either official test cases or human submissions with official verdicts. However, our method was specifically designed to generate test cases for problems that lack ground truth tests—a common situation since most problems in the wild do not have ground truth tests available. Consequently, ground truth verdicts are only available for a subset of problems in our dataset.
>
> Additionally, to enable fair comparison with baselines, we must evaluate on the intersection of our dataset with existing benchmarks such as TACO and CodeContests.
>
> The combination of these two constraints results in a small subset of 700 problems, among which we are using 650 for evaluation.
>
> **To provide more information, we provide precision evaluation on a larger subset of problems.** Note that this evaluation does not include comparison with baselines, because many of the problems we evaluate on are not included in the baselines.
>
> We selected all problems in HARDTESTS that originate from AtCoder, have public official test cases, and have a difficulty rating between 1 and 4, resulting in a total of 1.7K problems. We used Qwen2.5-Coder-7B-Instruct to generate five programs for each problem. The results are shown in the table below, and the performance is consistent with what we reported in the paper.
>
> | Metric                 | Result |
> | ---------------------- | ------ |
> | Difficulty 1 Precision | 97.58  |
> | Difficulty 1 Recall    | 98.91  |
> | Difficulty 2 Precision | 88.39  |
> | Difficulty 2 Recall    | 100.0  |
> | Difficulty 3 Precision | 69.66  |
> | Difficulty 3 Recall    | 88.57  |
> | Difficulty 4 Precision | 53.33  |
> | Difficulty 4 Recall    | 100.0  |
> | Average Precision      | 77.24  |
> | Average Recall         | 96.87  |
>
>
> ### W2: Verifier quality beyond correctness is not analyzed. the execution time of HARDTESTGEN’s test suites relative to prior work is not discussed ...  we would prefer a minimal test count that still accurately verifies correctness
>
> **We agree that minimizing test count and execution time are important considerations, and our results demonstrate that HardTestGen achieves both.**
>
> As detailed in Appendix A.10 (L2082-2085), each problem in HARDTESTS contains an average of **39 test cases**, compared to **109.6 test cases** in TACO, our primary baseline. Despite using **65% fewer tests**, HardTestGen achieves superior precision and recall, demonstrating that our method generates more efficient test suites.
>
> Regarding execution time, evaluating a candidate program on CPU takes only **5.1 seconds** on average (Appendix A.10). This overhead is negligible compared to the time required for LLM inference and weight updates during training, making our approach practical for real-world use.
>
> We recognize the importance of these efficiency metrics and will move them to the main body of the paper in our revision to give them appropriate prominence.

---

> ### Author Response · Authors · 2025-11-25
> **Response Part 2/3**
>
> ### W3: when a test suite is found to be incorrect (FP or FN), is that suite still retained in the final 26.6k-problem dataset? Even if improvements over prior work hold, precision in the 80% or 50–60% ranges (Figure 1) could inject noisy signals into post-training.
>
> **We retain all test suites in our dataset because ground truth verification is unavailable for most problems, and existing evidence suggests our quality threshold is sufficient for effective training.**
>
> As stated in the first part of our response, determining whether a test suite produces false positives or false negatives requires either official test cases or submissions to online judges, which are unavailable for most problems in our dataset. Our method was specifically designed to address this unavailability of official tests, so for the majority of problems, we cannot actually verify whether they might elicit false positives or negatives. Therefore, all generated test suites are retained in the final 26.6k-problem dataset.
>
> Regarding the concern about noise in post-training: existing baseline datasets such as TACO and CodeContests, despite having substantially lower quality than our data (as shown in Figure 1), have already been extensively and successfully used for training LLMs to code [1-5]. Our experiments demonstrate that HardTestGen not only achieves better precision than these widely-adopted baselines, but also produces superior downstream performance. HardTestGen is fundamentally a technique for producing better training data, and improvements in downstream model performance—which we demonstrate—are the ultimate criterion for success. This suggests that our quality threshold, while imperfect, is sufficient for effective training.
>
> ### W4: Cascaded LLM components lack evaluation ... a manually inspected subset could establish its reliability
>
> We manually checked the input generator, input validator, and output judging functions for 100 problems for correctness. We found that 95%, 94%, and 92% of them are correct respectively. These intrinsic evaluation will be included in our revision.
>
>
> ### Q1: how does the quality of the generated datasets vary with stronger frontier models (which have moved beyond GPT-4o in coding/reasoning)?
>
> As reported in **Table 7 of Appendix A.9**, we experimented with GPT-4o, Claude-4-Sonnet, Kimi-K2, and Qwen3-Coder as test case generators on 500 randomly selected AtCoder problems. Among these models, Claude-4-Sonnet is arguably the strongest. The results reflect that. Compared to GPT-4o, Claude-Sonnet-4 performs much better in test precision especially for difficulty 3 and 4.
>
> ### Q2: What percentage/number of test cases are filtered out? Have you tried automatic re-attempts (regeneration) to raise precision/recall when a generated suite is flagged as incorrect?
>
> As described in Section 3.4, each oracle program is executed on all synthesized inputs to produce outputs. Afterwards, if two oracle programs generate outputs that are equivalent on more than 90% of the inputs, only the matching inputs are preserved (for problems with only a single oracle program, we do not apply this filtering, and such cases account for only about 6% of the entire HARDTESTS dataset). Therefore, at most about 10% of valid inputs are filtered out for each problem. The table below shows the distribution of the number of test cases in HARDTESTS, and we can see that the vast majority of problems have a sufficiently large number of test cases.
>
> |Number of Test Cases|1 to 9|10 to 19|20 to 39|40 to 80|
> |---|---|---|---|---|
> |Percentage|1.7%|5.3%|51.8%|41.2%|

---

> ### Author Response · Authors · 2025-11-25
> **Response Part 3/3**
>
> ### Q3: Decontamination in 3.5. How exactly is decontamination performed (e.g., exact-match criteria, near-duplicate thresholds)?
>
> Thank you for your question! The detailed decontamination process is as follows. We will also put them into our paper.
>
> We collected problems from five direct data sources (TACO, CodeContests, Luogu, Codeforces, and AtCoder), so did LiveCodeBench.
>
> We than compare each problem in our dataset with every problem from LiveCodeBench, using the following criteria to check whether two problems are too similar:
>
> - If two problems come from different source platforms, we convert their problem statements into 3-gram sets and compute their Jaccard similarity (intersection over union). If the score exceeds 0.7, we treat them as the same problem and only keep one.
> - If two problems come from the same source platform, e.g. AtCoder, we instead compare their official problem IDs on that platform. We do not apply n-gram–based similarity in this case because problem setters on the same platform may slightly modify an existing problem to create a newer and harder version, which can make the n-gram similarity unreliable. In contrast, the platform-assigned problem IDs are much more reliable.
>
> ### Q4: Why are gains on human submissions smaller than on model-generated programs?
>
> A majority of these gains come from false positives, i.e., incorrect submissions being judged as correct.
>
> In that sense, human submissions are biased towards fewer obvious false positives, because human competition programmers usually check their solution with sample test cases and manually created test cases.
>
> It is less likely for a human submission to have easily spottable bugs, thus the number of false positives from human submissions than that from model generated ones.
>
> ### Q5: A more informative experiment would be similar to Figure 3 with an A/B setup for rejection sampling
>
> The purpose of our rejection sampling experiments is to motivate that having a good verifier matters for rejection sampling. We generate 5 trajectories for each questions in Hardtests and use Hardtests test cases to evaluate each of the generated trajectories. We only select questions where we have at least one correct and one incorrect trajectory (as determined by the test cases).
>
> Because of this setting, running an A/B setup using TACO verification feedback would be unfair: we cannot get passing trajectories with TACO test cases for the same 5000 problems as the HardTests ones.
>
> **Reference**
>
> [1] Xu, Shusheng, et al. "Is DPO superior to PPO for LLM alignment? a comprehensive study." Proceedings of the 41st International Conference on Machine Learning. 2024.
>
> [2] Ahmad, Wasi Uddin, et al. "OpenCodeReasoning: Advancing Data Distillation for Competitive Coding." Second Conference on Language Modeling, 2025.
>
> [3] Singh, Avi, et al. "Beyond Human Data: Scaling Self-Training for Problem-Solving with Language Models." Transactions on Machine Learning Research.
>
> [4] NovaSky Team. "Sky-T1: Train Your Own O1 Preview Model within $450." NovaSky, 2025, novasky-ai.github.io/posts/sky-t1.
>
> [5] Luo, Michael, et al. "DeepCoder: A Fully Open-Source 14B Coder at O3-mini Level." Notion Blog, 2025, pretty-radio-b75.notion.site/DeepCoder-A-Fully-Open-Source-14B-Coder-at-O3-mini-Level-1cf81902c14680b3bee5eb349a512a51.

---

### Author Response · Authors · 2025-12-03

We thank the reviewers for their engagement, feedback, and recognition of the contribution of our work, which addresses an important and underexplored gap in test-suite generation for training code LLMs. We are glad that the reviewers find our work **"well-motivated, novel", "impactful, valuable", "rigorous and broadly evaluated", "practical, scalable, and reproducible"**.

As the discussion period ended earlier than expected, we were unfortunately unable to engage further with the reviewers. We hope this summary clarifies our approach and demonstrates that we have carefully addressed the reviewers’ comments and concerns, most of which are already supported by the analyses and details provided in the appendix of our initial submission.

We summarized our response to the major concerns about our method and experiments below:

* ``Even if improvements over prior work hold, precision in the 80% or 50–60% ranges could inject noisy signals into post-training`` — We explained that all test suites are retained due to limited ground-truth verification, and that our quality threshold is already much better than existing training datasets and sufficient for effective post-training performance.
* ``Can HARDTESTGEN operate when no oracle exists?`` — As the reviewer has noted, we provided an “oracle-free” variant in **Appendix A.8**, showing the potential of HardTestGen for generalizing to when no oracle exists, although not perfect. In our original main text, we envisioned that developing better methods for synthesizing tests without an oracle would be one of the future directions.
* ``Is HARDTESTGEN cost-effective?`` — We argued that HardTestGen does not rely on expensive GPT-4o and can use significantly cheaper open-source models with similar performance (as shown in **Appendix A.9**). Therefore, scaling to millions of tasks does not introduce prohibitive cost.
* ``Verifier quality beyond correctness (cost profile) not analyzed`` — We referred to **Appendix A.10**, which details the test count and execution time of our method, demonstrating that HardTestGen achieves both minimizing test count and execution time.
* ``Cascaded LLM components lack evaluation`` — We manually checked 100 problems, finding 95%, 94%, and 92% correctness for the input generator, input validator, and output judging functions, respectively.

---

### Meta-Review · Area_Chair_ChKn · 2026-01-07

**Summary:**

Reviewers like how this is a well-motivated work that addresses an important gap in test-suite generation in an important area (Ksf8, UCch)
Reviewers like the modular design that capture diverse failure modes and find the pipeline using consensus sensible. They appreciated the rigorous evaluation and strength of the result (RVF6).

For weaknesses, reviewer Ksf8 raised how the estimate is based on a small subset, which is addressed by the author, due to the nature of using ground truth tests. Indeed in this case the subsample is not fully random, and may cause some issues. Reviewers also raised issues on the level of analysis and limited evaluations, all of which addressed by the authors.

Overall, the consensus is to accept. The authors should take reviewer feedback into account for their final version.

**Reviewer Concerns:**

most concerns are addressed, especially from reviewer Ksf8
* Precision estimates are based on a small subset
* Cascaded LLM components lack evaluation
* is it cost effective
* Verifier quality beyond correctness is not analyzed

**Reviewer Scores:**

N/A

---

### Decision · Program_Chairs · 2026-01-26

Accept (Poster)